# Quagmires in SFT-RL Post-Training: When High SFT Scores Mislead and What to Use Instead

**Feiyang Kang**[1,2,*,†]**, Michael Kuchnik**[1]**, Karthik Padthe**[1]**, Marin Vlastelica**[3]**, Ruoxi Jia**[2]**,
Carole-Jean Wu**[1]**, Newsha Ardalani**[1,†]

[1]FAIR at Meta, [2]Virginia Tech, [3]Meta

## Abstract

In post-training for reasoning Large Language Models (LLMs), the current state of practice trains LLMs in two independent stages: Supervised Fine-Tuning (SFT) and Reinforcement Learning with Verifiable Rewards (RLVR, shortened as "RL" below). In this work, we challenge whether high SFT scores translate to improved performance after RL. We provide extensive counter-examples where this is not true. We find high SFT scores can be biased toward simpler or more homogeneous data and are not reliably predictive of subsequent RL gains or scaled-up post-training effectiveness. In some cases, RL training on models with improved SFT performance could lead to substantially worse outcome compared to RL on the base model without SFT. We study alternative metrics and identify generalization loss on held-out reasoning examples and Pass@large k performance to provide strong proxies for the RL outcome. We trained hundreds of models up to 12B-parameter with SFT and RLVR via GRPO and ran extensive evaluations on 7 math benchmarks with up to 256 repetitions, spending >1M GPU hours. Experiments include models from Llama3, Mistral-Nemo, Qwen3 and multiple state-of-the-art SFT/RL datasets. Compared to directly predicting from pre-RL performance, prediction based on generalization loss and Pass@large k achieves substantial higher precision, improving $R^2$ coefficient and Spearman's rank correlation coefficient by up to 0.5 (2x). This provides strong utility for broad use cases. For example, in most experiments, we find SFT training on unique examples for a one epoch underperforms training on half examples for two epochs, either after SFT or SFT-then-RL; With the same SFT budget, training only on short examples may lead to better SFT performance, though, it often leads to worse outcomes after RL compared to training on examples with varying lengths. This work develops an enhanced evaluation tool for math reasoning tasks and is open-sourced[1].

## 1 Introduction

The evolution of Large Language Models (LLMs) has seen a significant focus on enhancing their reasoning abilities, a process heavily reliant on post-training (Wen et al., 2025). This phase refines pre-trained models, adapting them for complex, multi-step tasks like mathematics, logic, and code generation, leading to the emergence of Large Reasoning Models (LRMs) (Kumar et al., 2025). The open-sourced DeepSeek R1 achieved phenomenal success in pushing forward the frontier of LLM's reasoning capabilities (Guo et al., 2025). Its new post-training paradigm, Reinforcement Learning with Verifiable Rewards (RLVR) via Group Relative Policy Optimization (GRPO) (Liu et al., 2024; Shao et al., 2024), has shown substantial improvements on top of previous post-training methods. Following DeepSeek R1's practice, current works typically conduct SFT before RL, assuming models with better performance after SFT will ultimately be better after RL (Liu et al., 2025b; Wen et al., 2025). In industrial practice, these post-training stages are often distributed among different teams, with SFT and RL handled by separate groups, each optimizing for their own performance metrics (Chen et al., 2025b; Meta, 2025). This process relies on the intuition that a model with stronger SFT

---

[*]Work done at Meta.[†]Correspondence to: Feiyang Kang <fyk@vt.edu>, Newsha Ardalani <new@meta.com>

[1]Code repo: `https://github.com/feiyang-k/SFT-RL-ICLR`

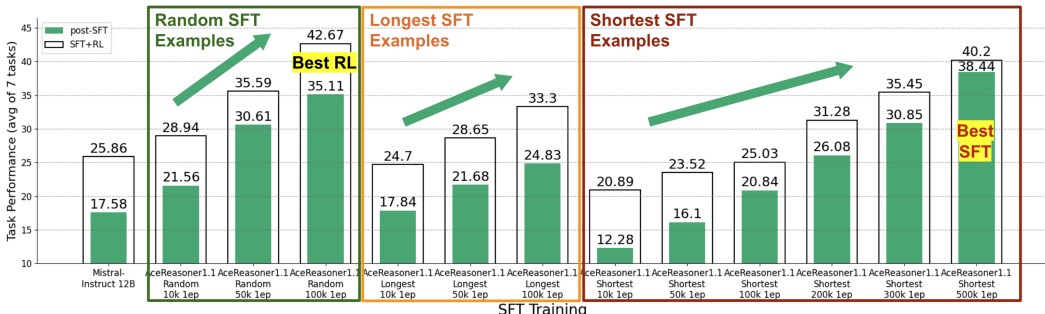

Figure 1: Mistral-NeMo-12B-Instruct undergone SFT-RL with SFT examples from AceReasoner1.1-SFT dataset and RLVR via GRPO on DeepScaleR dataset. Reporting Pass@1 performance averaged over 7 math benchmarks. When training on Random/Longest/Shortest SFT examples, *the final performance after RL increases at different rates than the SFT performance.* Model with the best SFT performance is not the one with the best final performance after RL. Post-SFT and SFT+RL performance correlate, though, optimizing post-SFT performance might not optimize the final performance after RL.

performance will yield better outcomes after RLVR (Liu et al., 2025b). With efforts and resources being poured in improving post-training paradigms and data recipes, also escalating are the debates on *whether SFT helps or hurts the subsequent RL training*.

In this setup, post-training strategies and data are often designed either for SFT or RL, but not jointly. In practice, SFT and RL are often conducted sequentially (e.g., Rastogi et al. (2025)). SFT data is usually selected to maximize evaluation performance after SFT (Zhang et al., 2025; Ye et al., 2025), and the best-performing SFT models are believed to also yield stronger performance after subsequent RL. However, this assumption is often flawed. Over-training during SFT, for instance, can constrain the model's behavior and limit the exploration crucial for effective RL (Chen et al., 2025a; Wang et al., 2025). For example, we find training on repeated examples for up to 8 epochs leads to better SFT performance than training on the same data for 2 epochs (4x compute) but yields visibly worse outcome after RL (Figure 4, left). On the contrary, Cen et al. (2025) shows SFT training on manually crafted "exploratory" examples, despite leading to a lower performance after SFT, helps achieve better final outcome after RL. This leads to a critical gap in the current practice:

*An SFT-trained model with the best evaluation performance may not be the best candidate for subsequent training with RLVR (e.g., Figure 1).*

When the final RLVR performance is unsatisfactory, it becomes challenging to attribute the failure to either the RL stage or a non-ideal SFT starting point. This misalignment can cause friction and overhead between teams. Furthermore, the high computational cost of RL training and long pipelines, especially in agentic use cases, makes end-to-end tuning across the SFT-RL stages prohibitively expensive (Toledo et al., 2025). Early stopping during RL is also generally ineffective, as the model with the fastest initial improvement may not achieve the highest final performance (Liu et al., 2025b). Even with identical post-training procedures, different models may respond vastly different (Figure 2). Consequently, a significant gap remains in our ability to reliably predict RLVR outcomes.

This work centers on addressing this predictability problem. We ask the following research questions:

RQ1: *Do models with better pre-RL performance always lead to better outcomes after RLVR? If not, what are the failure modes? (Section 3)*

RQ2: *What are effective SFT paradigms and data recipes when considering subsequent RLVR training? Can we determine the suitability of an SFT model before committing to the expensive RL stage? (Section 4)*

To tackle these questions, we first examine the relationship between pre-RL performance and post-RL outcomes across various SFT training paradigms and data recipes. While we often observe some extents of correlation between post-SFT performance and final outcome after RL, we identify cases where the trends visibly diverge. For instance, training on the same dataset for more epochs may significantly boost post-SFT performance but diminishes the potential for improvement during RL,

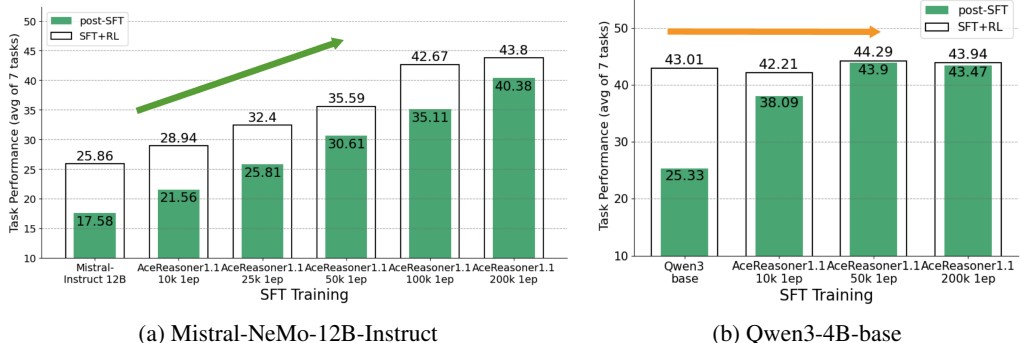

(a) Mistral-NeMo-12B-Instruct      (b) Qwen3-4B-base

Figure 2: Both models undergone SFT-RL with SFT examples from AceReasoner1.1-SFT dataset and RLVR via GRPO on DeepScaleR dataset. Reporting Pass@1 performance averaged over 7 math benchmarks. *Even with identical post-training procedures, different models may respond vastly different.* With increasing SFT examples, Mistral's (left) post-SFT performance and final performance after RL both increase. Yet, for Qwen3 models (right), the post-SFT performances appear uncorrelated with the final performance after RL, where the latter remains the same despite the substantially improved SFT performance.

sometimes resulting in lower final performance (e.g., Figure 4). Similarly, training on simpler, shorter reasoning examples improves pre-RL performance quickly, but these models gain much less from the subsequent RL stage. These findings are particularly concerning given that many SFT data selection methods favor simpler or more homogeneous examples (Zhang et al., 2025; Yu et al., 2025).

Next, we identify more reliable predictors for RL success. We find that as SFT proceeds, an eventual increase in validation loss is strongly correlated with performance improvements in the later RL stage. Furthermore, since the RL objective is to compress Pass@k performance into Pass@1 (i.e., maximize expected reward), we investigate using Pass@k at a large k as a predictor. We conduct extensive empirical validation using Llama3-8B (Grattafiori et al., 2024), Mistral-Nemo-12B (team, 2024), and Qwen3-4B-base (Yang et al., 2025) models on state-of-the-art SFT datasets like Llama-Nemotron (Singhal et al.) and AceReasoner1.1 (Liu et al., 2025b) and different RL datasets. Our results demonstrate that these new metrics can reliably predict the outcome of RLVR, improving $R^2$ coefficient and Spearman's rank correlation coefficient by up to 0.5 (2x), providing strong utility for broad use cases. For example, in most experiments, we find SFT training on unique examples for a one epoch underperforms training on half examples for two epochs, either after SFT or SFT-then-RL; With the same SFT budget, training only on short examples may lead to better SFT performance, though, it often leads to worse outcome after RL compared to training on examples with varying lengths. These can be captured by the proposed predictors, but not from the post-SFT performance.

To address the limitations in available tools, we developed an enhanced tool for more convenient and reliable evaluation of reasoning models, which will be open-sourced in contribution to the community.

## 2 RELATED WORKS

The research landscape for reasoning post-training and data strategies is fast evolving and in its early days. In a typical setup, post-training for reasoning LMs conducts SFT and RL sequentially, which has been reported to work better than only conducting SFT or RL (Rastogi et al., 2025). Viewpoints in many impactful works can be inconsistent or even contradicting: "Initial 'cold-start' SFT is necessary for subsequent RL" (DeepSeek-R1 technical report, Guo et al. (2025)); "over-SFT may constrain subsequent RL" (Llama-4 technical report, Meta (2025)); "SFT generalizes poorly and RL without SFT does better" (Chen et al., 2025a), showing prominent gaps in characterizing post-training dynamics and the role of each stage. The lack of predictability in the post-training outcome poses a major blocker for optimizing training paradigms or data recipes.

### 2.1 POST-TRAINING FOR REASONING: SFT-THEN-RLVR PARADIGM

Post-training for reasoning LLMs typically consists of two or three stages: a) **Supervised Fine-Tuning (SFT)**, b) an optional **Direct Preference Optimization (DPO)** stage, and c) **Reinforcement**

**Learning with Verifiable Rewards (RLVR)** (Lambert et al., 2024). SFT serves as the "cold-start" phase, providing the model with a strong initial policy by exposing it to high-quality reasoning chains (Guo et al., 2025). The model is trained on problems with high-quality solutions sourced from the frontier models. DPO fixes/strengthens targeted behaviors (e.g., precise instruction following in math/logic derivations) assessed important for effective reasoning, which is more subjective and often optional (Lambert et al., 2024). RL further improves the model's reasoning and problem-solving capability. This allows the model to explore the solution space more broadly than SFT alone, discovering novel and more robust reasoning paths.

While the sequential SFT-then-RL pipeline is dominant, researchers have explored alternative paradigms to more tightly integrate or unify these learning stages. Efforts include iterate or interleave SFT and RL (Meta, 2025), gradually shift from SFT to RL while increasing task difficulty (Yang et al., 2025), or directly unify the objectives of SFT and RL (Xu et al., 2025). Though these are promising research directions, they come with their own complexities and have not yet universally replaced the SFT-then-RL paradigm, which remains a robust and widely-adopted industry standard. Many important issues regarding the stability, data requirements, and effectiveness of these unified methods remain to be solved. Our work, therefore, focuses on improving the predictability and efficiency of the prevailing SFT-then-RL pipeline, providing practical tools that are immediately applicable to current state-of-the-art workflows.

## 2.2 RECENT ADVANCEMENTS AND CURRENT CHALLENGES

In post-training for reasoning, SFT data is usually selected to maximize evaluation performance after SFT (Li et al., 2025; Ye et al., 2025), and the best-performing SFT models are believed to also yield stronger performance after subsequent RL. Significant research effort is now focused on more sophisticated selection and curation strategies for SFT data. Techniques range from filtering for complexity and diversity to generating synthetic data that covers a wider range of reasoning structures (Rastogi et al., 2025; Yuan et al., 2025; Ye et al., 2025; Abdin et al., 2024). Some methods propose selecting data points based on their difficulty and influence, aiming to find a subset of examples that provides the strongest learning signal (Muennighoff et al., 2025). Current efforts prioritize scaling up SFT training on existing models, leading to new SOTA performance on reasoning tasks for those models (Guha et al., 2025). A significant challenge is that standard SFT performance metrics, such as average accuracy on benchmarks, are not always predictive of post-RL success. This creates a critical gap between the optimization target of the SFT stage and the final performance of the model.

Several issues contribute to this gap. First, models can overfit to the specific patterns and artifacts present in the SFT dataset, leading to poor generalization during the exploration phase of RL (Chen et al., 2025a). Furthermore, naively collecting or generating data can lead to datasets that lack diversity in reasoning strategies or are skewed toward simpler problems, causing the SFT-trained model to develop biases that stifle exploration in the subsequent RL stage (Guha et al., 2025). The landscape is further fogged by the recently reported data contamination issues (Wu et al., 2025). The results from these models have served as the basis for many research findings.

The (lack of) predictability for final performance after RL from pre-RL models leads to quagmires for post-training. SFT teams may provide suboptimal RL learners. It creates frictions between post-training teams owning different SFT and RL stages and chaos in optimizing the training paradigm/data recipes, adding overheads on the model development and hindering productivity. It calls for new tools that better characterize the post-training dynamics and predictive of the RL outcome. This will have profound impact on broad downstream fields–research and applications alike–from improving SFT data curation, search for the next post-training paradigm, to RL for non-verifiable tasks, etc.

## 3 THE SFT METRIC TRAP

Previous works, from SFT data selection to RL training methodologies, have often operated under a common assumption. They *implicitly* assume or *explicitly* argue that models exhibiting better post-SFT performance will consistently yield superior final outcomes after subsequent reinforcement learning (Rastogi et al., 2025; Liu et al., 2025b). This assumption has justified the widespread practice of optimizing the SFT and RL stages in isolation, with teams or processes focusing on maximizing SFT evaluation metrics as a primary goal. However, the separation of SFT and RL optimization can

lead to a widening gap in reasoning post-training, where improvements in the initial stage do not translate to the final stage. This motivates us to ask two fundamental questions:

- *Do models with better pre-RL performance always lead to better outcomes after RLVR?*

- *If not, what are the failure modes?*

To investigate these questions, we design experiments across two representative scenarios that reflect common practices and research directions in the field: a "dataset-level" analysis and an "instance-level" analysis. In **Dataset-Level Scenarios**, SFT examples are drawn from the same data distribution, but we vary the amount of unique samples and the training paradigm (e.g., learning rate, number of epochs); In **Instance-Level Scenarios**, we consider training on different datasets while keeping the training pipeline fixed (i.e., using the same model and training paradigm). This setup is primarily concerned with SFT data selection and curation, examining whether strong SFT performance on a given dataset transfers to the final outcome after RL.

## 3.1 DATASET-LEVEL SCENARIOS

In this scenario, we draw SFT examples from the same underlying data distribution but vary the training configuration, such as the number of unique samples/training epochs/learning rate. This setup is highly relevant to industrial practices where SFT and RL are often handled by different teams. In current practices, the number of training epochs is a design choice often determined by practical factors such as data availability or compute budget. Specifically, when the amount of training samples is a more prominent constraint (such as domains with limited high-quality examples), repeating for more epochs on the data may be preferred to improve post-SFT performance. On the contrary, if data is abundant relative to the allocated compute budget (for this domain/capability), current practices (such as Singhal et al.) may prefer to train for just a single epoch on unique examples.

In these cases, the training paradigm is determined **heuristically** where the only optimizable target is the post-SFT performance. Surprisingly, we identified both practices to be **suboptimal**. We found that post-SFT performance often improves stably when training for more epochs–even with excessive overtraining. But models overtrained during SFT show decreasing potentials for the subsequent RL. Typically, the model with the best final performance after RL is not the one with the best post-SFT performance. Further, with the same compute budget for SFT, training on more data for one epoch typically leads to visibly lower post-SFT performance compared to training on less data for a few more epochs, and the final performance after RL remains underperforming. A concrete example is provided in Figure 4. High SFT scores can be biased toward *homogeneous or repeated examples* and are not reliably predictive of subsequent RL gains.

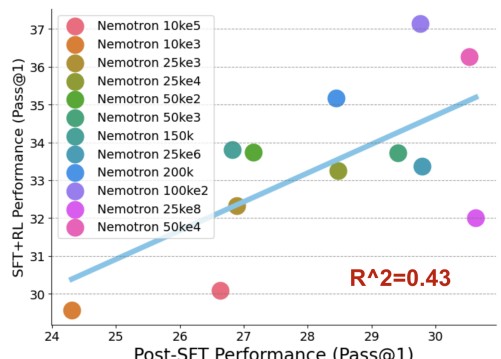

Figure 3: Llama3-8B-Instruct undergone SFT-RL with SFT examples from Llama-Nemotron-SFT dataset and RLVR via GRPO on MATH dataset (train-split). Reporting Pass@1 performance averaged over 7 math benchmarks. Linear fit between post-SFT performance and final outcome after RL. The two performance correlates with $R^2 = 0.43$, indicating *post-SFT performance explains only 43% of variation in the final outcome after RL and the remaining gaps are prominent.*

This mismatch between post-SFT and post-RL performance is not directly visible from post-SFT models. As shown in Figure 3 where we fit a linear function between post-SFT and post-RL performance, these two performance correlates with $R^2 = 0.43$, indicating post-SFT performance only explains 43% of variation in the final outcome after RL whereas the gaps remain evident.

## 3.2 INSTANCE-LEVEL SCENARIOS

In this scenario, we fix the model and the training configurations but vary the SFT datasets. This setup is primarily concerned with SFT data selection and curation, examining whether the strong

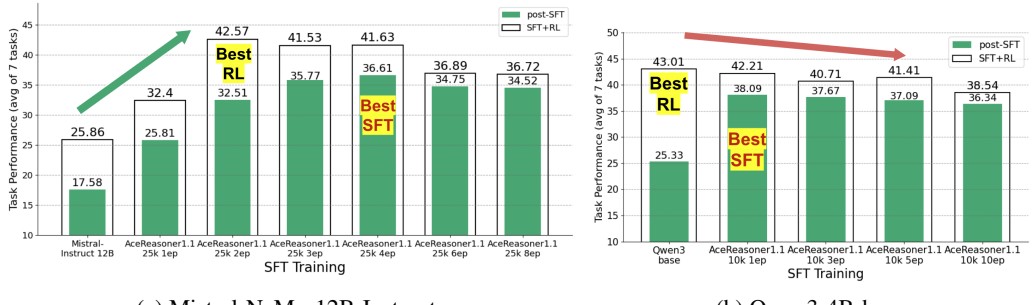

(a) Mistral-NeMo-12B-Instruct     (b) Qwen3-4B-base

Figure 4: Both models undergone SFT-RL with SFT examples from AceReasoner1.1-SFT dataset and RLVR via GRPO on DeepScaleR dataset. Reporting Pass@1 performance averaged over 7 math benchmarks. When repeating SFT for more epochs on the same data, Mistral's (left) SFT continues to improve with up to 4 epochs where the final performance after RL saturates after 2 epochs. Qwen3's (right) final performance after RL degrades with SFT training, though, these models' post-SFT performance is substantially higher than the base model. *Both cases show clear divergence between post-SFT performance and final performance after RL. Here, optimizing post-SFT performance will be suboptimal or ineffective for improving the final model.*

SFT performance promised by a particular dataset transfers to strong final performance after RL. For instance, state-of-the-art data selection methods are often prone to selecting examples that are more "natural" or easier for the model to learn (Zhang et al., 2025; Yu et al., 2025). While this simpler data may allow the model to achieve high SFT metrics more quickly, we question whether this comes at the cost of learning more difficult or advanced reasoning capabilities that are crucial for downstream success. We identified similar gaps between post-SFT performance and final outcome after RL. Visualizing representative examples in Figure 1, high SFT scores can be biased toward *simpler examples* and are not reliably predictive of subsequent RL gains or scaled-up post-training effectiveness. For example, *training on shortest examples led to faster performance improvements* than training on randomly sampled examples during SFT. These shorter examples are closer to the model's original generations and easier to learn, though, these are not best examples for the model to gain reasoning capabilities in preparation for RL. The final performance after RL is significantly lower. These gaps are not directly captured in the post-SFT performance.

# 4 PROPOSED METRICS TOWARDS MORE RELIABLE PREDICTIONS

## 4.1 GENERALIZATION LOSS ON VALIDATION EXAMPLES

During the investigation above, we identified a counterintuitive pattern in which post-SFT performance improves stably when training for more epochs whereas the overtrained models show decreased potentials during the subsequent RL. The best final performance after RL is not usually achieved on models with the best post-SFT performance. To be able to optimize the final outcome on the given training examples, one needs to optimize the SFT training paradigm based on the predicted final outcome after RL. We materialize this insight and identify generalization loss after SFT to be a viable indicator of the model's potential during the subsequent RL. While repeating training for more epochs, together with the improving post-SFT performance, we observe the generalization loss on validation examples to elevate and eventually flare up, indicating strong over-fitting. This generalization loss shows strong correlation with further performance gains during subsequent RL, allowing prediction for the final outcome after RL (Figure 5). When using it in practice, after conducting SFT training with different numbers of examples and epochs, we can immediately rule out post-SFT models with both lower performance and higher generalization loss as they will likely remain underperforming after the subsequent RL, facilitating determination of the best SFT training paradigm.

## 4.2 PASS@K ACCURACY EVALUATED AT LARGE K

The objective of RLVR via GRPO is to maximize expected reward, which explicitly optimizes the Pass@1 accuracy on the RL tasks. GRPO only progresses when at least one of the responses for the RL task is correct. Recent works argue that GRPO compresses Pass@k accuracy into Pass@1 (Yue et al., 2025), and empirical evidence appears to support the argument showing GRPO mostly improves

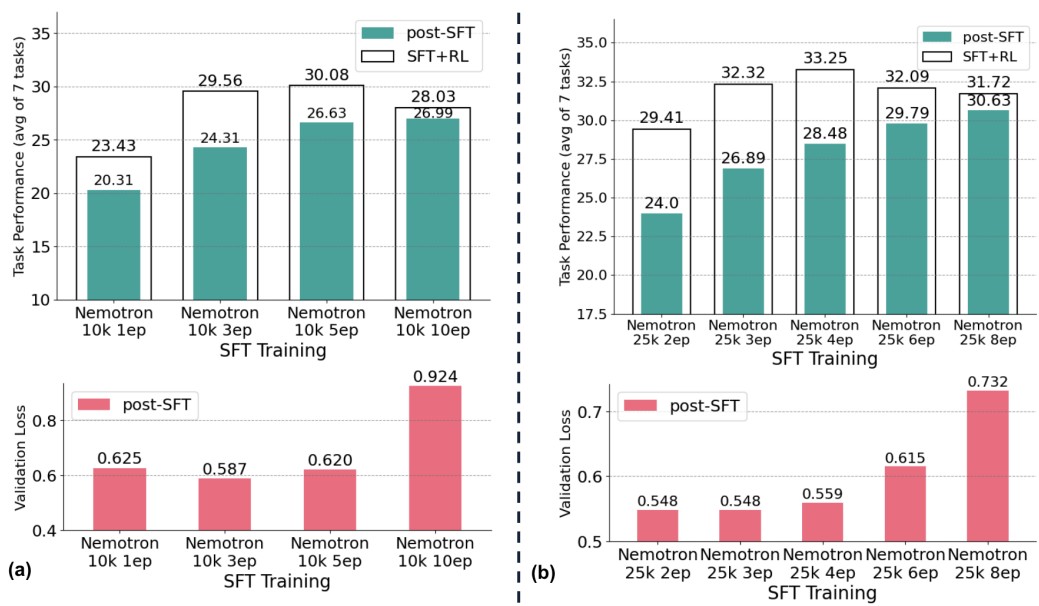

Figure 5: Llama3-8B-Instruct undergone SFT-RL with SFT examples from Llama-Nemotron-SFT dataset and RLVR via GRPO on MATH dataset (train-split). Reporting Pass@1 performance averaged over 7 math benchmarks and generalization loss on the validation set of SFT data. We identify generalization loss after SFT to be a viable indicator for the model's RL potential. While repeating training for more epochs, together with the improving post-SFT performance, we observe the generalization loss on validation examples to elevate and eventually flare up, indicating strong over-fitting. *This generalization loss shows strong correlation with the further performance gain during the subsequent RL, allowing prediction for the final outcome after RL.*

average Pass@1 accuracy on tasks where the original model achieves an above-zero accuracy (Liu et al., 2025b). Though it remains debatable whether GRPO discovers new solution traces beyond the capabilities of the original model (Liu et al., 2025a), all these analyses and findings suggest RLVR dynamics during GRPO training to be strongly coupled with the original models Pass@k accuracy. Hu et al. (2023) pioneers in using the Pass@high metric to study the scaling of task performance. The authors argue that Pass@k provides finer resolution to the Pass@1 metric and better captures the underlying dynamics. Acting on this intuition, we consider Pass@k performance of the post-SFT model, especially with large k, as a candidate metric for predicting its final outcome after the subsequent RL. When using it in practice, after SFT training, we evaluate Pass@k performance on the post-SFT models with different values of k. For efficient implementation, we leverage the following formula which provides unbiased estimations for Pass@k accuracies for all $k \leq n$ (Brown et al., 2024),

$$\text{Pass@}k = \mathbb{E}\left(1 - \frac{\binom{n-c}{k}}{\binom{n}{k}}\right) \tag{1}$$

where integer $n$ denotes the total number of responses generated for the task, integer $k$ denotes the target value for k Pass@k, and integer $c$ denotes the number of correct responses for the task, respectively.

We consider the Pass@large k performance as the indicator for the final outcome after RL and deem the post-SFT model with the best Pass@large k performance to have the best Pass@1 performance after RL. The best post-SFT model can be determined without needing to conduct any actual RL run.

## 5 EMPIRICAL EVALUATIONS

### 5.1 SETUP

We conduct three sets of experiments with SFT-RL post-training. On **Llama3-8B-Instruct** models, we conduct SFT training with examples from Llama-Nemotron dataset (where we only select math

samples with responses generated by QwQ-32B (Team, 2025) or DeepSeek R1 (Guo et al., 2025), hereinafter the same) and RL training on MATH dataset (train-split) (Hendrycks et al., 2021); on **Mistral-Nemo-12B-Instruct** and Qwen3-4B-base, we conduct SFT training with examples from AceReasoner1.1-SFT dataset and RL training on DeepScaleR dataset (Luo et al., 2025). For all models, we conduct RL training for 3 epochs where each run takes up to 5 days. We repeat RL training for 4+ runs on each data recipe and training paradigm, conduct 4+ evaluations on different checkpoints across RL training run, and report the best performance for the model. We evaluate task performance on 7 math benchmarks, **MATH-500** (Hendrycks et al., 2021), **AIME 1983-2024** (Veeraboina, 2023), **GSM8k** (Cobbe et al., 2021), **AIME 2025** (of America, 2025), **AMC** (Competitions, 2025), **Olympiad** (He et al., 2024), **Minerva** (Lewkowycz et al., 2022), and report model performance as Pass@1 accuracy averaged over 64 repetitions and across 7 tasks. For the proposed predictors, we evaluate the generalization loss on the validation set of the SFT data and Pass@64 accuracy averaged over 256 repetitions. Experiments spent >1M GPU hours on NVIDIA A100. Please refer to Appendix B for additional details. Shown in Figure 2 (right), in this setup, Qwen3-series models do not appear to benefit from state-of-the-art SFT datasets, and models undergone different SFT training achieve considerably close final performance after RL. Since this work focuses on studying the impact of different SFT training on the subsequent RL, we present these results as qualitative examples instead (deferred to Appendix A).

Following the categorization above, we organize experiments in two major scenarios: **dataset-level** prediction, and **instance-level** prediction. In dataset-level prediction experiments: we conduct SFT training for the base model on samples from math reasoning datasets with different training paradigms (varying number of examples and epochs). In instance-level prediction experiments: we first create diverse different curated SFT datasets by selecting the shortest/longest subsets, random samples, or their different mixtures (samples are shown in Appendix E). Then, we conduct SFT training for the base model on samples from each curated dataset with the same training paradigms (one epoch). We consider two primary metrics measure prediction performance, **Coefficient of determination** ($R^2$) (Pearson, 1909), and **Spearman's rank correlation coefficient (Spearman)** (Zar, 1972). Specifically, $R^2$ measures the proportion of variation in the prediction variable (final performance) that is unexplained by the predictor, examining the accuracy of prediction on the final performance after RL. Spearman yields a number ranging from -1 to 1 that indicates how strongly two sets of ranks are correlated, which we use to examine the effectiveness in identifying post-SFT models that lead to the best final performance. Additional results can be found in Appendix C.

## 5.2 USE CASE 1: DATASET-LEVEL PREDICTION

This use case focuses on optimizing the SFT training paradigm, a common dataset-level challenge. Given a fixed compute budget, practitioners must decide on the optimal trade-off between the volume of unique data and the number of training epochs, navigating the risks of under- and over-training. We test the predictive power of our proposed metrics against the baseline of using post-SFT Pass@1 accuracy. To examine the accuracy of prediction with $R^2$, we randomly select 50% SFT models and fit a linear function between their post-SFT performance and final performance after RL. The fitted function is then used to predict the final performance of the other 50% SFT models. We compare the predictions to their actual post-RL outcomes to compute $R^2$. We repeat the random sampling for 100 times and report the standard error.

Table 1: Spearman's rank correlation between performance predicted from post-SFT models and the actual performance after RL. Both generalization loss and Pass@64 achieve notable margins over prediction from Pass@1, whereas averging the two prediction may or may not lead to better results.

| Spearman's Rank Correlation / Models | Prediction based on SFT Pass@1 (avg. of 64) baseline | Prediction based on SFT Generalization Loss | Prediction based on SFT Pass@Large k (k=64) | Avg. Prediction from SFT Gen. Loss + Pass@Large k (64) |
|---|---|---|---|---|
| Llama3-8B-Instruct | 0.75 | 0.94 | 0.95 | **0.97** (+0.22) |
| Mistral-NeMo-12B-Instruct | 0.78 | 0.90 | **0.92** (+0.14) | 0.90 |

Table 2: Measuring prediction accuracy with coefficient of determination ($R^2$). We randomly select 50% SFT models and fit a linear function between their post-SFT performance and performance after RL, and use it to predict for the other 50% SFT models. We repeat random sampling for 100 times and report standard errors.

| Coefficient of determination ($R^2$) / Models | Prediction based on SFT Pass@1 (avg. of 64) baseline | Prediction based on SFT Generalization Loss | Prediction based on SFT Pass@Large k (k=64) | Avg. Prediction from SFT Gen. Loss + Pass@Large k (64) |
|---|---|---|---|---|
| Llama3-8B-Instruct | $0.57 \pm 0.29$ | $0.88 \pm 0.09$ | $0.87 \pm 0.10$ | $\mathbf{0.94} \pm 0.04$ (+0.37) |
| Mistral-NeMo-12B-Instruct | $0.29 \pm 0.38$ | $\mathbf{0.79} \pm 0.26$ (+0.50) | $0.57 \pm 0.32$ | $0.72 \pm 0.24$ |

---

**Takeaway 1: Dataset-level Prediction**

- Both generalization loss and Pass@large k are effective predictors for post-RL performance when optimizing SFT training configurations on a single dataset, providing higher-accuracy estimates that help guide decisions and save significant compute.

- Both predictors excel at identifying correct rankings for post-RL performance, achieving $\geq$ 0.90 Spearman correlation (30% improvements); generalization loss provides advantageous prediction accuracy ($R^2$) for post-RL performance with up to 2x improvements.

---

## 5.3 USE CASE 2: INSTANCE-LEVEL PREDICTION

This use case addresses the challenge of SFT data selection, an instance-level optimization problem. Here, the training pipeline is fixed, but we aim to select the optimal SFT dataset from a pool of candidates curated with different strategies (e.g., selecting for shortest/longest solutions, diversity, etc. Ye et al. (2025)). This scenario tests whether strong SFT performance on a given dataset translates to a good final outcome after RL.

Table 3: Spearman's rank correlation between performance predicted from post-SFT models and the actual performance after RL. Pass@64 achieve notable margins over prediction from Pass@1.

| Spearman's Rank/ Correlation / Models | Prediction based on SFT Pass@1 (avg. of 64) baseline | Prediction based on SFT Pass@Large k (k=64) |
|---|---|---|
| Llama3-8B-Instruct | 0.69 | $\mathbf{0.94}$ (+0.25) |
| Mistral-NeMo-12B-Instruct | 0.70 | $\mathbf{0.98}$ (+0.28) |

Table 4: Measuring prediction accuracy with coefficient of determination ($R^2$). We randomly select 50% SFT models and fit a linear function between their post-SFT performance and performance after RL, and use it to predict for the other 50% SFT models. We repeat random sampling for 100 times and report standard errors.

| Coefficient of determination ($R^2$) / Models | Prediction based on SFT Pass@1 (avg. of 64) baseline | Prediction based on SFT Pass@Large k (k=64) |
|---|---|---|
| Llama3-8B-Instruct | $0.58 \pm 0.20$ | $\mathbf{0.92} \pm 0.05$ (+0.34) |
| Mistral-NeMo-12B-Instruct | $0.73 \pm 0.16$ | $\mathbf{0.98} \pm 0.01$ (+0.25) |

*In this scenario, the generalization loss predictor is not applicable.* Since each SFT dataset comes from a different distribution, the validation loss includes a distributional gap component in addition to generalization error. Without a common, representative validation set, it is difficult to make a fair comparison. Pass@large k metric proves to be exceptionally robust. Since it measures the model's inherent capability to produce correct solutions, it is less sensitive to distributional shifts in the training data. It can be used to effectively rank different SFT datasets and select the one with the highest potential for RL, without needing to run any RL experiments for calibration.

---

**Takeaway 2: Instance-level Prediction**

- Pass@large k turns out highly accurate and robust in instance-level predictions, improving Spearman correlation by up to 36% and prediction accuracy ($R^2$) by up to 59%. It effectively identifies datasets for strong post-RL performance and predicts RL outcomes.

- Generalization loss is not applicable for instance-level selection due to distributional gaps between different datasets.

---

**How to use them in practice?**  Our metrics support two primary workflows. If the goal is simply to *rank* SFT candidates, one can use generalization loss to quickly filter out clearly suboptimal models (i.e., those with both low performance and high loss). Then, Pass@large k can be used to reliably rank the remaining candidates to identify the most promising one. If the goal is to *predict the final performance value*—for instance, to inform trade-offs between SFT costs and expected gains—practitioners can run RL on a small number of SFT models to gather calibration data. A linear predictor can then be fitted using our proposed metrics, allowing for accurate performance estimation across all SFT candidates without the need for exhaustive RL runs.

## 6   CONCLUSIONS

This work confronts a critical quagmire in reasoning post-training: the common assumption that high SFT scores guarantee strong performance after subsequent RL. Through extensive experimentation with Llama3/Mistral-Nemo/Qwen3 models spending >1M GPU hours, we provide broad counter-examples where SFT performance is often misleading or biased toward simpler/repeated data. Our primary contribution is the identification and validation of two more reliable predictors for post-RL success: generalization loss on held-out reasoning examples and Pass@large k accuracy, improving prediction accuracy ($R^2$) and Spearman's rank correlation by up to 0.5 (2x) over prediction from post-SFT performance. By allowing practitioners to better predict the final outcome, our work helps de-risk the expensive RL stage and streamline the entire post-training pipeline. We open-sourced our enhanced evaluation tool to facilitate broader adoption and further research. This work focuses on mathematical reasoning. A natural **next step** is to study the topic in a wider range of reasoning tasks (e.g., coding, science) and agentic use cases; Our study is limited to the prevailing paradigm of online RL with GRPO. The relationship between SFT characteristics and post-RL performance with other methods such as offline RL/DPO or other RL algorithms may worth further explorations; Directly evaluating Pass@large k requires repeating evaluation for at least k times, which becomes computational expensive with long sequence lengths. Estimating Pass@k accuracy from that of smaller k holds the promise for more efficient evaluations (Schaeffer et al., 2025).

## 7   DISCUSSIONS

Broader investigations and supplementary examples are provided in Appendix D. In Appendix D.1, we conduct fine-grained comparisons between the best SFT model vs. the best SFT+RL model. *We show extensive examples that the best overall models (SFT+RL) achieve notable performance margins in the final performance (SFT+RL) over the best SFT models, despite trailing in SFT performance.* To facilitate the pursuit towards underlying mechanism behind the intriguing SFT-RL dynamics, in Appendix D.2, we investigate the aggregated entropy at the response level. As shown in Tables 15, 16, and 17, we witnessed that correct responses generally show a lower entropy compared to incorrect responses. However, we observe *1. A lower entropy after SFT training does not always suggest worse post-RL outcomes, and vice versa. 2. A lower entropy after SFT training does not indicate better post-SFT performance, nor does the entropy gap between correct and incorrect responses.*

We found that during decoding, math/symbolic tokens generally show a near-zero loss–i.e., the model is almost always certain on these math tokens. The highest loss tokens are human languages that are naturally ambiguous or interchangeable, such as whether to start the sentence with "Thus" or "Therefore". As a result, the (per-token) entropy for the generations mostly measures *verbosity*, and does not show consistent patterns with accuracy. In general, we did not observe a consistent pattern to explain the SFT-RL dynamic, but it confirms that *entropy alone is an inadequate explanation*.

## ACKNOWLEDGMENTS

Ruoxi Jia and the ReDS lab acknowledge support from the National Science Foundation through grants IIS-2312794, IIS-2313130, OAC-2239622. The authors extend their sincere appreciation to the reviewers, Area Chairs, and the organizing committee for their dedication throughout the review process. Amid unprecedented challenges, they demonstrated exceptional commitment to upholding the integrity and quality of the proceedings, ensuring the timely dissemination of this work to the community.

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

# Appendices

## A  ADDITIONAL SFT-RL EXAMPLES AND VISUALIZATIONS

### A.1  LLAMA3-8B-INSTRUCT

Figure 6 shows results on Llama3-8B-Instruct undergone SFT-RL with SFT examples from Llama-Nemotron-SFT/AceReasoner1.1-SFT/OpenR1-Math (Hugging Face, 2025) dataset and RLVR via GRPO on MATH dataset (train-split). Reporting average Pass@1 performance on MATH-500 (test-split). *High SFT scores can be biased toward simpler or more homogeneous data and are not reliably predictive of subsequent RL gains or post-training effectiveness.* SFT on fewer unique examples repeated for more training epochs (ep) or/and with a larger learning rate (LR) leads to higher accuracy on reasoning benchmarks such as MATH-500 (+8.75% vs. non-repeated data, left figure). However, models trained this way show smaller improvements during RL (-1.43% vs. non-repeated). In contrast, SFT on more diverse, non-repeated data—despite yielding lower initial SFT performance (-5% vs. repeated data, middle/right figure)—results in significantly better post-RL performance (+5.94%).

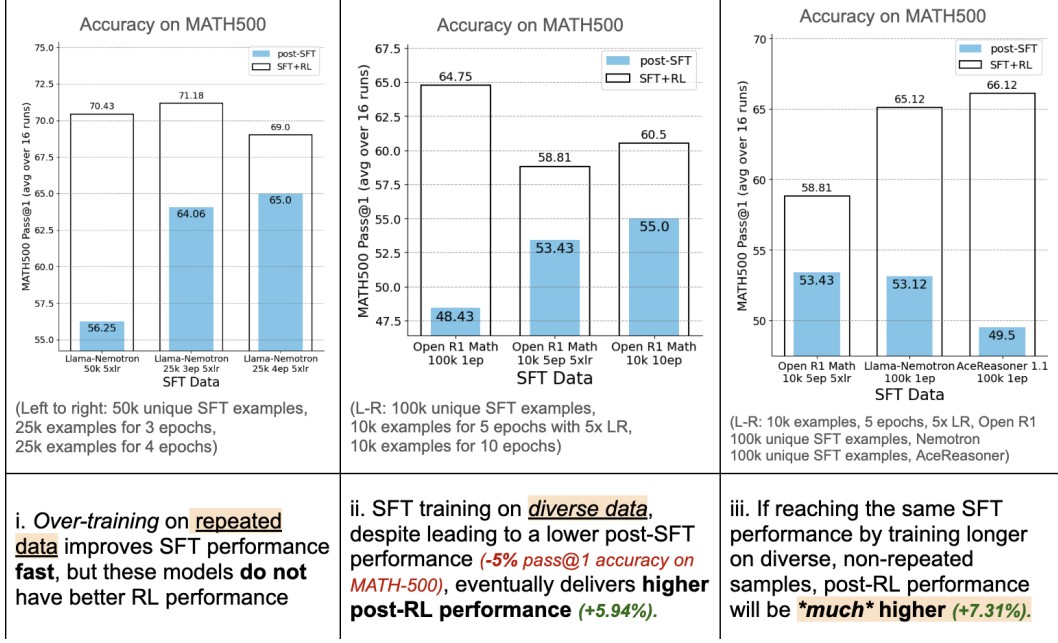

Figure 6:  Llama3-8B-Instruct undergone SFT-RL with SFT examples from Llama-Nemotron-SFT/AceReasoner1.1-SFT/OpenR1-Math dataset and RLVR via GRPO on MATH dataset (train-split). Reporting average Pass@1 performance on MATH-500 (test-split). *High SFT scores can be biased toward simpler or more homogeneous data and are not reliably predictive of subsequent RL gains or post-training effectiveness.* SFT on fewer unique examples repeated for more training epochs (ep) or/and with a larger learning rate (LR) leads to higher accuracy on reasoning benchmarks such as MATH-500 (+8.75% vs. non-repeated data, left figure). However, models trained this way show smaller improvements during RL (-1.43% vs. non-repeated). In contrast, SFT on more diverse, non-repeated data—despite yielding lower initial SFT performance (-5% vs. repeated data, middle/right figure)—results in significantly better post-RL performance (+5.94%).

Figure 7 shows results on Llama3-8B-Instruct undergone SFT-RL with SFT examples from Llama-Nemotron-SFT dataset and RLVR via GRPO on MATH dataset (train-split). Reporting Pass@1 performance averaged over 7 math benchmarks. High SFT scores can be biased toward *simpler examples* and are not reliably predictive of subsequent RL gains or scaled-up post-training effectiveness. For example, training on shortest examples (e.g., s10k, s500k) led to faster performance improvements than training on randomly sampled examples (e.g., 10k, 200k) during SFT (lower smaller dots). These shorter examples are closer to the model's original generations and easier to learn, though, these are not best examples for the model to gain reasoning capabilities in preparation for RL. The final performance after RL (upper larger dots) is significantly lower.

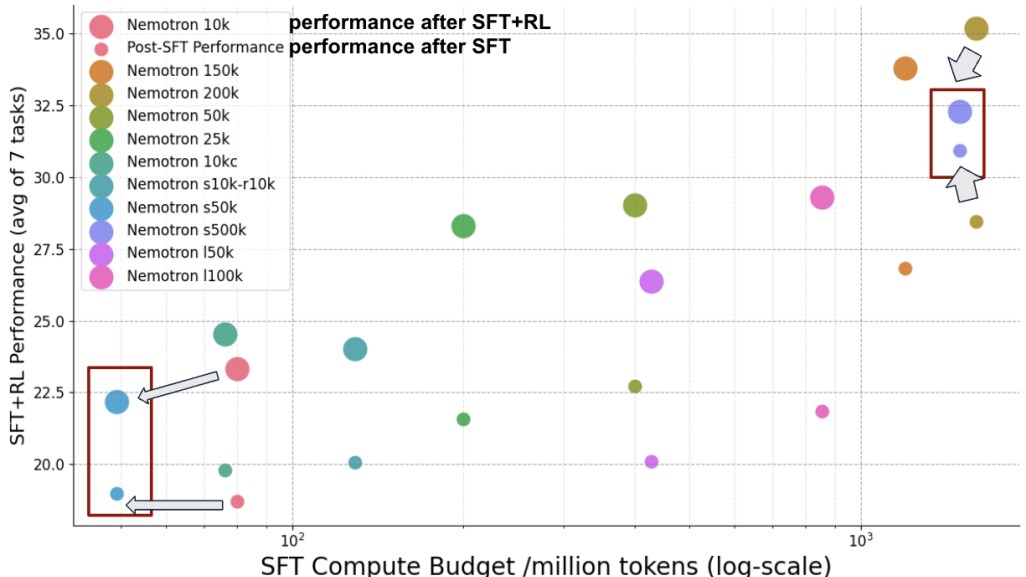

Figure 7: Llama3-8B-Instruct undergone SFT-RL with SFT examples from Llama-Nemotron-SFT dataset and RLVR via GRPO on MATH dataset (train-split). Reporting Pass@1 performance averaged over 7 math benchmarks. High SFT scores can be biased toward *simpler examples* and are not reliably predictive of subsequent RL gains or scaled-up post-training effectiveness. For example, training on shortest examples (e.g., s10k, s500k) led to faster performance improvements than training on randomly sampled examples (e.g., 10k, 200k) during SFT (lower smaller dots). These shorter examples are closer to the model's original generations and easier to learn, though, these are not best examples for the model to gain reasoning capabilities in preparation for RL. The final performance after RL (upper larger dots) is significantly lower.

## A.2 MISTRAL-NEMO-12B-INSTRUCT

Figure 8 shows results on Mistral-NeMo-12B-Instruct undergone SFT-RL with **shortest** SFT examples from AceReasoner1.1-SFT dataset and RLVR via GRPO on DeepScaleR dataset. Reporting Pass@1 performance averaged over 7 math benchmarks. With increasing SFT examples, Mistral's post-SFT performance first dips and then gradually recovers and improves to performance better than before SFT training. Compared to the base model, the final performance after RL also first dips and then gradually goes up and improves to a better level. *Notably, post-RL performance recovers to the same level as the base model slower than the post-SFT performance.* The post-SFT and post-RL performance trends are not identical.

## A.3 QWEN3-4B-BASE

Figure 9 shows results on Qwen3-4B-base undergone SFT-RL with **shortest** SFT examples from AceReasoner1.1-SFT dataset and RLVR via GRPO on DeepScaleR dataset. Reporting Pass@1 performance averaged over 7 math benchmarks. With increasing SFT examples, Qwen3's post-SFT performances appear uncorrelated with the final performance after RL, where the latter remains the same despite the substantially improved SFT performance.

Figure 12 shows results on Qwen3-4B-base undergone SFT-RL with **Shortest/Longest/-Longest+Shortest** SFT examples from AceReasoner1.1-SFT dataset and RLVR via GRPO on DeepScaleR dataset. Reporting Pass@1 performance averaged over 7 math benchmarks. All SFT training substantially improves Qwen3's post-SFT performance, but the final performance after RL is mixed. Training on **Longest** and 10k **Longest**+10k **Shortest** SFT examples lead to visibly improved final performance after RL where the latter achieves the best final performance for Qwen3 models in this work. Other SFT training lead to significantly degraded final performance after RL.

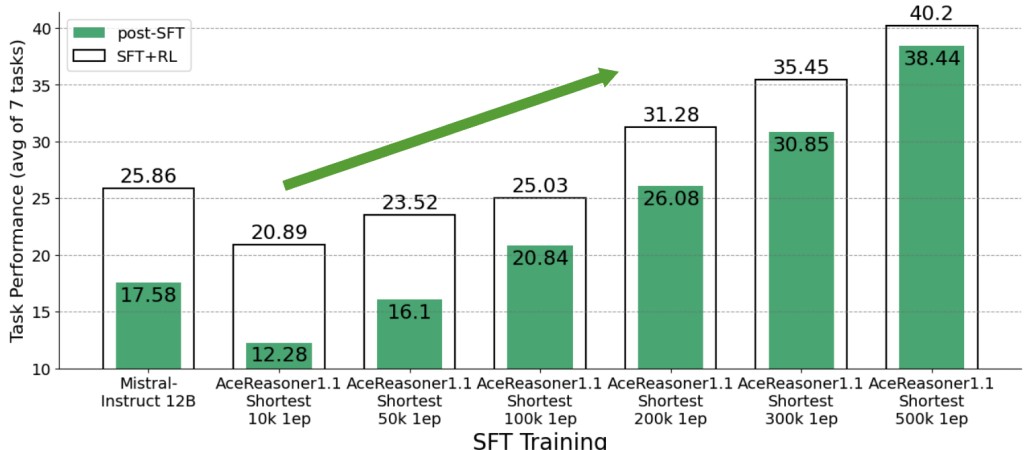

Figure 8: Mistral-NeMo-12B-Instruct undergone SFT-RL with **shortest** SFT examples from AceReasoner1.1-SFT dataset and RLVR via GRPO on DeepScaleR dataset. Reporting Pass@1 performance averaged over 7 math benchmarks. With increasing SFT examples, Mistral's post-SFT performance first dips and then gradually recovers and improves to performance better than before SFT training. Compared to the base model, the final performance after RL also first dips and then gradually goes up and improves to a better level. *Notably, post-RL performance recovers to the same level as the base model slower than the post-SFT performance.* The post-SFT and post-RL performance trends are not identical.

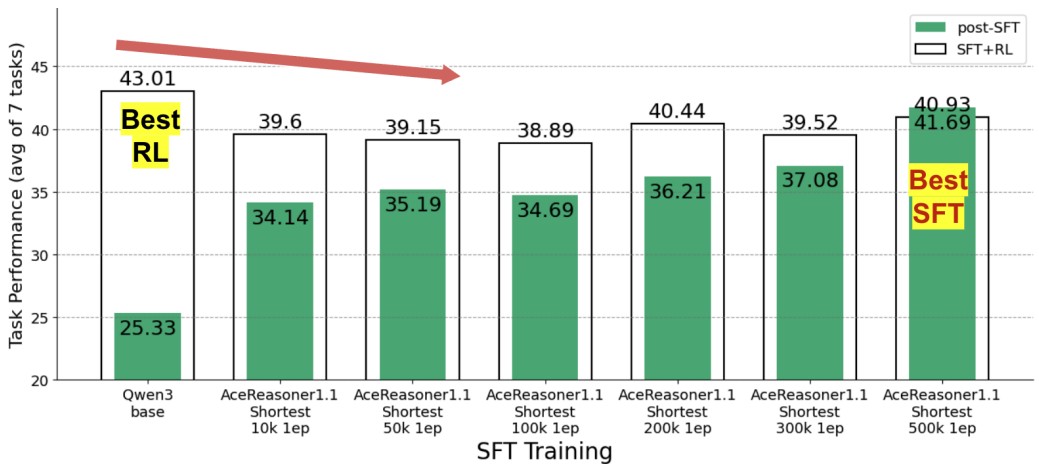

Figure 9: Qwen3-4B-base undergone SFT-RL with **shortest** SFT examples from AceReasoner1.1-SFT dataset and RLVR via GRPO on DeepScaleR dataset. Reporting Pass@1 performance averaged over 7 math benchmarks. With increasing SFT examples, Qwen3's post-SFT performances appear uncorrelated with the final performance after RL, where the latter remains the same despite the substantially improved SFT performance.

## B  IMPLEMENTATION DETAILS

All experiments (SFT, RL, evaluation) are conducted on individual AWS (Mathew & Varia, 2014) node with 8x NVIDIA A100 80GB GPU. Experiments spent >1M GPU hours on NVIDIA A100 80GB. We repeat RL training for 4+ runs on each data recipe and training paradigm (each run takes up to 5 days), conduct 4+ evaluations on different checkpoints across RL training run, and report the best performance for the model. We set the max sequence length to 8k tokens throughout SFT, RL, and evaluation.

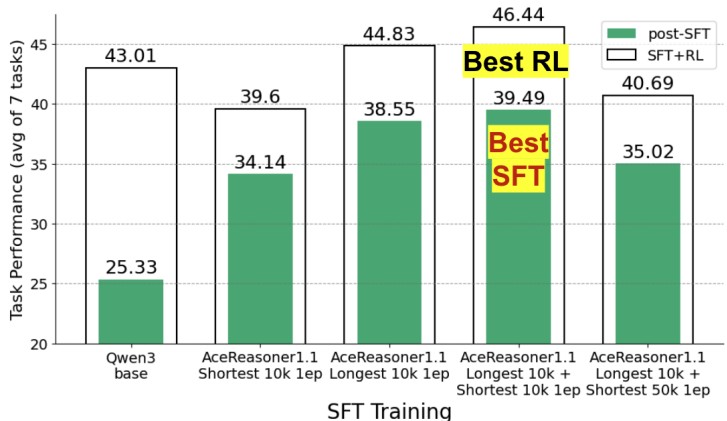

Figure 10: Qwen3-4B-base undergone SFT-RL with **shortest/Longest/Longest+Shortest** SFT examples from AceReasoner1.1-SFT dataset and RLVR via GRPO on DeepScaleR dataset. Reporting Pass@1 performance averaged over 7 math benchmarks. All SFT training substantially improves Qwen3's post-SFT performance, but the final performance after RL is mixed. Training on **Longest** and 10k **Longest**+10k **Shortest** SFT examples lead to visibly improved final performance after RL where the latter achieves the best final performance for Qwen3 models in this work. Other SFT training lead to significantly degraded final performance after RL.

### B.1  MODELS AND DATASETS

We conduct three sets of experiments with SFT-RL post-training. On **Llama3-8B-Instruct** models, we conduct SFT training with examples from Llama-Nemotron dataset (where we only select math samples with responses generated by QwQ-32B (Team, 2025) or DeepSeek R1 (Guo et al., 2025), hereinafter the same) and RL training on MATH dataset (train-split) (Hendrycks et al., 2021); on **Mistral-Nemo-12B-Instruct** and Qwen3-4B-base, we conduct SFT training with examples from AceReasoner1.1-SFT dataset and RL training on DeepScaleR dataset (Luo et al., 2025). For all models, we conduct RL training for 3 epochs where each run takes up to 5 days.

### B.2  TRAINING

We conduct SFT training with LLaMA-Factory (Zheng et al., 2024) using learning rates lr=1e-5 and global batch size = 128, and RL training via GRPO with verl (Sheng et al., 2024) using learning rates lr=1e-6 and global batch size = 128. We sample 16 rollouts for each question with temperature=1.0. We set KL loss coefficient=0 and entropy coefficient=0.001.

### B.3  EVALUATION

Evaluations are conducted with pipelines originally developed in this work based on vllm (Kwon et al., 2023) and HuggingFace's math-verify (Kydlíček, 2025), enabling efficient inference with performant and accurate verification. We ran evaluations with the same template and generation configuration as in RL, using decoding temperature t=1.0 and the standard reasoning prompt (`"Let's think step by step and output the final answer within \\boxed{}."`).

We evaluate task performance on 7 math benchmarks, including **MATH-500** (Hendrycks et al., 2021), **AIME 1983-2024** (Veeraboina, 2023), **GSM8k** (Cobbe et al., 2021), **AIME 2025** (of America, 2025), **AMC** (Competitions, 2025), **Olympiad** (He et al., 2024), **Minerva** (Lewkowycz et al., 2022), and report model performance as Pass@1 averaged over 64 repetitions and across 7 tasks. For the proposed predictors, we evaluate the generalization loss on the validation set of the SFT data and Pass@64 accuracy averaged over 256 repetitions.

## C  ADDITIONAL EXPERIMENTAL RESULTS

### C.1  DATASET-LEVEL

Table 5 shows results on Llama3-8B-Instruct undergone SFT-RL with SFT examples from Llama-Nemotron-SFT dataset and RLVR via GRPO on MATH dataset (train-split). Reporting Pass@1 performance averaged over 7 math benchmarks. Measuring prediction accuracy with coefficient of determination ($R^2$) varying the ratio of fit-validation datapoints. We randomly select x SFT models and fit a linear function between their post-SFT performance and performance after RL, and use it to predict for the rest SFT models. We repeat random sampling for 100 times and report standard errors.

Table 6 shows results on Mistral-NeMo-12B-Instruct undergone SFT-RL with SFT examples from AceReasoner1.1-SFT dataset and RLVR via GRPO on DeepScaleR dataset. Measuring prediction accuracy with coefficient of determination ($R^2$) varying the ratio of fit-validation datapoints. Reporting Pass@1 performance averaged over 7 math benchmarks. We randomly select x SFT models and fit a linear function between their post-SFT performance and performance after RL, and use it to predict for the rest SFT models. We repeat random sampling for 100 times and report standard errors.

Table 5: Llama3-8B-Instruct undergone SFT-RL with SFT examples from Llama-Nemotron-SFT dataset and RLVR via GRPO on MATH dataset (train-split). Reporting Pass@1 performance averaged over 7 math benchmarks. Measuring prediction accuracy with coefficient of determination ($R^2$) varying the ratio of fit-validation datapoints. We randomly select x SFT models and fit a linear function between their post-SFT performance and performance after RL, and use it to predict for the rest SFT models. We repeat random sampling for 100 times and report standard errors.

| No. of Fitting-Validation Datapoints / Coefficient of determination ($R^2$) | Prediction based on SFT Pass@1 (avg. of 64) baseline | Prediction based on SFT Generalization Loss | Prediction based on SFT Pass@Large k (k=64) | Avg. Prediction from SFT Gen. Loss + Pass@Large k (64) |
|---|---|---|---|---|
| Fitting: 3; Validation: 13 | $0.48 \pm 0.40$ | $0.80 \pm 0.33$ | $0.80 \pm 0.23$ | $\mathbf{0.86} \pm 0.22 (+0.38)$ |
| Fitting: 4; Validation: 12 | $0.57 \pm 0.29$ | $0.82 \pm 0.21$ | $0.84 \pm 0.15$ | $\mathbf{0.92} \pm 0.08 (+0.35)$ |
| Fitting: 5; Validation: 11 | $0.57 \pm 0.29$ | $0.88 \pm 0.09$ | $0.87 \pm 0.10$ | $\mathbf{0.94} \pm 0.04 (+0.37)$ |
| Fitting: 6; Validation: 10 | $0.57 \pm 0.26$ | $0.89 \pm 0.07$ | $0.87 \pm 0.10$ | $\mathbf{0.95} \pm 0.03 (+0.38)$ |
| Fitting: 7; Validation: 9 | $0.64 \pm 0.19$ | $0.89 \pm 0.07$ | $0.90 \pm 0.06$ | $\mathbf{0.95} \pm 0.05 (+0.31)$ |
| Fitting: 8; Validation: 8 | $0.64 \pm 0.20$ | $0.88 \pm 0.08$ | $0.88 \pm 0.08$ | $\mathbf{0.93} \pm 0.05 (+0.29)$ |
| Fitting: 10; Validation: 6 | $0.59 \pm 0.33$ | $0.85 \pm 0.17$ | $0.85 \pm 0.15$ | $\mathbf{0.94} \pm 0.05 (+0.35)$ |
| Fitting: 12; Validation: 4 | $0.54 \pm 0.43$ | $0.86 \pm 0.18$ | $0.81 \pm 0.23$ | $\mathbf{0.91} \pm 0.12 (+0.37)$ |

Table 6: Mistral-NeMo-12B-Instruct undergone SFT-RL with SFT examples from AceReasoner1.1-SFT dataset and RLVR via GRPO on DeepScaleR dataset. Measuring prediction accuracy with coefficient of determination ($R^2$) varying the ratio of fit-validation datapoints. Reporting Pass@1 performance averaged over 7 math benchmarks. We randomly select x SFT models and fit a linear function between their post-SFT performance and performance after RL, and use it to predict for the rest SFT models. We repeat random sampling for 100 times and report standard errors.

| No. of Fitting-Validation Datapoints / Coefficient of determination ($R^2$) | Prediction based on SFT Pass@1 (avg. of 64) baseline | Prediction based on SFT Generalization Loss | Prediction based on SFT Pass@Large k (k=64) | Avg. Prediction from SFT Gen. Loss + Pass@Large k (64) |
|---|---|---|---|---|
| Fitting: 3; Validation: 7 | $0.32 \pm 0.39$ | $\mathbf{0.73} \pm 0.41 (+0.41)$ | $0.52 \pm 0.31$ | $0.61 \pm 0.38$ |
| Fitting: 4; Validation: 6 | $0.27 \pm 0.36$ | $\mathbf{0.75} \pm 0.34 (+0.48)$ | $0.51 \pm 0.37$ | $0.69 \pm 0.26$ |
| Fitting: 5; Validation: 5 | $0.29 \pm 0.38$ | $\mathbf{0.79} \pm 0.26 (+0.50)$ | $0.57 \pm 0.32$ | $0.72 \pm 0.24$ |
| Fitting: 6; Validation: 4 | $0.37 \pm 0.37$ | $\mathbf{0.78} \pm 0.25 (+0.41)$ | $0.57 \pm 0.37$ | $0.67 \pm 0.35$ |
| Fitting: 7; Validation: 3 | $0.36 \pm 0.36$ | $\mathbf{0.77} \pm 0.30 (+0.41)$ | $0.57 \pm 0.35$ | $0.66 \pm 0.37$ |
| Fitting: 8; Validation: 2 | $0.31 \pm 0.46$ | $\mathbf{0.68} \pm 0.36 (+0.37)$ | $0.47 \pm 0.54$ | $0.64 \pm 0.37$ |

### C.2  INSTANCE-LEVEL

Table 7 shows results on Llama3-8B-Instruct undergone SFT-RL with SFT examples from Llama-Nemotron-SFT dataset and RLVR via GRPO on MATH dataset (train-split). Reporting Pass@1 performance averaged over 7 math benchmarks. Measuring prediction accuracy with coefficient of determination ($R^2$) varying the ratio of fit-validation datapoints. We randomly select x SFT models

and fit a linear function between their post-SFT performance and performance after RL, and use it to predict for the rest SFT models. We repeat random sampling for 100 times and report standard errors.

Table 8 shows results on Llama3-8B-Instruct undergone SFT-RL with SFT examples from Llama-Nemotron-SFT dataset and RLVR via GRPO on MATH dataset (train-split). Reporting Pass@1 performance averaged over 7 math benchmarks. Spearman's rank correlation between performance predicted from post-SFT models and the actual performance after RL, grouped by different SFT training budget.

Table 7: Llama3-8B-Instruct undergone SFT-RL with SFT examples from Llama-Nemotron-SFT dataset and RLVR via GRPO on MATH dataset (train-split). Reporting Pass@1 performance averaged over 7 math benchmarks. Measuring prediction accuracy with coefficient of determination ($R^2$) varying the ratio of fit-validation datapoints. We randomly select x SFT models and fit a linear function between their post-SFT performance and performance after RL, and use it to predict for the rest SFT models. We repeat random sampling for 100 times and report standard errors.

| No. of Fitting-Validation Datapoints/ Coefficient of determination ($R^2$) | Prediction based on SFT Pass@1 (avg. of 64) baseline | Prediction based on SFT Pass@Large k (k=64) |
|---|---|---|
| Fitting: 3; Validation: 14 | $0.40_{\pm 0.31}$ | $\mathbf{0.89}_{\pm 0.10}$ (+0.49) |
| Fitting: 4; Validation: 13 | $0.49_{\pm 0.30}$ | $\mathbf{0.89}_{\pm 0.17}$ (+0.40) |
| Fitting: 5; Validation: 12 | $0.55_{\pm 0.22}$ | $\mathbf{0.91}_{\pm 0.05}$ (+0.36) |
| Fitting: 6; Validation: 11 | $0.54_{\pm 0.30}$ | $\mathbf{0.92}_{\pm 0.04}$ (+0.38) |
| Fitting: 7; Validation: 10 | $0.55_{\pm 0.24}$ | $\mathbf{0.92}_{\pm 0.04}$ (+0.37) |
| Fitting: 8; Validation: 9 | $0.58_{\pm 0.20}$ | $\mathbf{0.92}_{\pm 0.05}$ (+0.34) |
| Fitting: 10; Validation: 7 | $0.56_{\pm 0.25}$ | $\mathbf{0.92}_{\pm 0.05}$ (+0.36) |
| Fitting: 12; Validation: 5 | $0.57_{\pm 0.28}$ | $\mathbf{0.92}_{\pm 0.05}$ (+0.35) |

Table 8: Llama3-8B-Instruct undergone SFT-RL with SFT examples from Llama-Nemotron-SFT dataset and RLVR via GRPO on MATH dataset (train-split). Reporting Pass@1 performance averaged over 7 math benchmarks. Spearman's rank correlation between performance predicted from post-SFT models and the actual performance after RL, grouped by different SFT training budget.

| SFT Compute Budget/ Spearman's Rank Correlation | Prediction based on SFT Pass@1 (avg. of 64) baseline | Prediction based on SFT Pass@Large k (k=64) |
|---|---|---|
| Low Budget ($< 2B$ tokens) | 0.77 | $\mathbf{0.99}$ (+0.22) |
| Medium Budget ($2 \sim 5B$ tokens) | 0.60 | $\mathbf{0.90}$ (+0.30) |
| High Budget ($5 \sim 20B$ tokens) | 0.70 | $\mathbf{0.94}$ (+0.24) |
| Average | 0.69 | $\mathbf{0.94}$ (+0.25) |

Table 9 shows results on Mistral-NeMo-12B-Instruct undergone SFT-RL with SFT examples from AceReasoner1.1-SFT dataset and RLVR via GRPO on DeepScaleR dataset. Measuring prediction accuracy with coefficient of determination ($R^2$) varying the ratio of fit-validation datapoints. Mistral-NeMo-12B-Instruct undergone SFT-RL with SFT examples from AceReasoner1.1-SFT dataset and RLVR via GRPO on DeepScaleR dataset. Reporting Pass@1 performance averaged over 7 math benchmarks. We randomly select x SFT models and fit a linear function between their post-SFT performance and performance after RL, and use it to predict for the rest SFT models. We repeat random sampling for 100 times and report standard errors.

Table 10 shows results on Mistral-NeMo-12B-Instruct undergone SFT-RL with SFT examples from AceReasoner1.1-SFT dataset and RLVR via GRPO on DeepScaleR dataset. Reporting Pass@1 performance averaged over 7 math benchmarks. Spearman's rank correlation between performance predicted from post-SFT models and the actual performance after RL, grouped by different SFT training budget.

Table 9: Mistral-NeMo-12B-Instruct undergone SFT-RL with SFT examples from AceReasoner1.1-SFT dataset and RLVR via GRPO on DeepScaleR dataset. Measuring prediction accuracy with coefficient of determination ($R^2$) varying the ratio of fit-validation datapoints. Mistral-NeMo-12B-Instruct undergone SFT-RL with SFT examples from AceReasoner1.1-SFT dataset and RLVR via GRPO on DeepScaleR dataset. Reporting Pass@1 performance averaged over 7 math benchmarks. We randomly select x SFT models and fit a linear function between their post-SFT performance and performance after RL, and use it to predict for the rest SFT models. We repeat random sampling for 100 times and report standard errors.

| No. of Fitting-Validation Datapoints/ Coefficient of determination ($R^2$) | Prediction based on SFT Pass@1 (avg. of 64) baseline | Prediction based on SFT Pass@Large k (k=64) |
|---|---|---|
| Fitting: 2; Validation: 10 | $0.55_{\pm 0.42}$ | $\mathbf{0.87}_{\pm 0.29}$ (+0.32) |
| Fitting: 3; Validation: 9 | $0.71_{\pm 0.15}$ | $\mathbf{0.94}_{\pm 0.18}$ (+0.23) |
| Fitting: 4; Validation: 8 | $0.69_{\pm 0.22}$ | $\mathbf{0.98}_{\pm 0.03}$ (+0.29) |
| Fitting: 5; Validation: 7 | $0.75_{\pm 0.10}$ | $\mathbf{0.98}_{\pm 0.01}$ (+0.23) |
| Fitting: 6; Validation: 6 | $0.73_{\pm 0.16}$ | $\mathbf{0.98}_{\pm 0.01}$ (+0.25) |
| Fitting: 8; Validation: 4 | $0.69_{\pm 0.35}$ | $\mathbf{0.97}_{\pm 0.03}$ (+0.28) |
| Fitting: 10; Validation: 2 | $0.68_{\pm 0.42}$ | $\mathbf{0.91}_{\pm 0.17}$ (+0.23) |

Table 10: Mistral-NeMo-12B-Instruct undergone SFT-RL with SFT examples from AceReasoner1.1-SFT dataset and RLVR via GRPO on DeepScaleR dataset. Reporting Pass@1 performance averaged over 7 math benchmarks. Spearman's rank correlation between performance predicted from post-SFT models and the actual performance after RL, grouped by different SFT training budget.

| SFT Compute Budget/ Spearman's Rank Correlation | Prediction based on SFT Pass@1 (avg. of 64) baseline | Prediction based on SFT Pass@Large k (k=64) |
|---|---|---|
| Low Budget ($< 2B$ tokens) | 0.80 | $\mathbf{0.95}$ (+0.25) |
| Medium Budget ($2 \sim 5B$ tokens) | 0.80 | $\mathbf{1.00}$ (+0.20) |
| High Budget ($5 \sim 20B$ tokens) | 0.50 | $\mathbf{1.00}$ (+0.50) |
| Average | 0.70 | $\mathbf{0.98}$ (+0.28) |

# D  BROADER INVESTIGATIONS AND SUPPLEMENTARY EXAMPLES

## D.1  FINE-GRAINED COMPARISONS: BEST SFT MODEL VS. BEST SFT+RL MODEL

Individual task performance largely correlates with the average performance, but the relative improvements differ across tasks. The average performance over multiple tasks has been reported to scale much more smoothly to the point of becoming predictable (Gadre et al., 2024). Thus, in this work, we use the average performance across multiple tasks as the main metric. The final results are consistent with these expectations.

Table 11: Detailed comparison of post-SFT and SFT+RL performance (Pass@1 accuracy) between the model with the best SFT performance (M2=Mistral-NeMo-12B-Instruct with SFT on 500k *shortest* AceReasoner1.1-SFT examples for 1 epochs) and the model with the best SFT+RL performance (M1=Mistral-NeMo-12B-Instruct with SFT on 100k *random* AceReasoner1.1-SFT examples for 1 epochs) from Figure 4(a) across various benchmarks. *Despite M1 trailing M2 by 8.66% after SFT, in the final SFT+RL results, M1 leads M2 by a relative margin of +6.14% with up to +33.31% on individual task.*

| Model / Benchmark | Avg | MATH-500 | AIME 1983-2024 | GSM8k | AIME 2025 | AMC | Olympiad | Minerva |
|---|---|---|---|---|---|---|---|---|
| *Supervised Fine-Tuning (SFT)* | | | | | | | | |
| M1 SFT | **35.11** | 70.50 | 10.62 | 89.34 | 13.48 | 38.71 | 15.15 | 8.03 |
| M2 SFT | **38.44** | 76.68 | 16.31 | 87.71 | 14.53 | 46.34 | 18.56 | 8.97 |
| *(M1-M2)/M2 (%)* | *-8.66* | *-8.06* | *-34.89* | *+1.86* | *-7.23* | *-16.47* | *-18.37* | *-10.45* |
| *Supervised Fine-Tuning + Reinforcement Learning (SFT+RL)* | | | | | | | | |
| M1 SFT+RL | **42.67** | 79.15 | 25.90 | 85.43 | 19.73 | 54.56 | 23.53 | 10.40 |
| M2 SFT+RL | **40.20** | 79.90 | 23.71 | 86.96 | 18.33 | 44.62 | 17.65 | 10.25 |
| *(M1-M2)/M2 (%)* | *+6.14* | *-0.94* | *+9.24* | *-1.76* | *+7.64* | *+22.28* | *+33.31* | *+1.46* |

Table 12: Detailed comparison of post-SFT and SFT+RL performance (Pass@1 accuracy) between the model with the best SFT performance (M2=Mistral-NeMo-12B-Instruct with SFT on 25k AceReasoner1.1-SFT examples for 4 epochs) and the model with the best SFT+RL performance (M1=Mistral-NeMo-12B-Instruct with SFT on 25k AceReasoner1.1-SFT examples for 2 epochs) from Figure 1 across various benchmarks. *Despite M1 trailing M2 by 11.20% after SFT, in the final SFT+RL results, M1 leads M2 by a relative margin of +2.26% with up to +95.01% on individual task.*

| Model / Benchmark | Avg | MATH-500 | AIME 1983-2024 | GSM8k | AIME 2025 | AMC | Olympiad | Minerva |
|---|---|---|---|---|---|---|---|---|
| *Supervised Fine-Tuning (SFT)* | | | | | | | | |
| M1 SFT | **32.51** | 63.53 | 8.19 | 91.96 | 6.77 | 29.90 | 17.00 | 10.28 |
| M2 SFT | **36.61** | 69.34 | 12.12 | 91.78 | 14.43 | 37.37 | 20.71 | 10.28 |
| *(M1-M2)/M2 (%)* | *-11.20* | *-8.38* | *-32.45* | *-0.20* | *-53.08* | *-19.99* | *-17.91* | *0.00* |
| *Supervised Fine-Tuning + Reinforcement Learning (SFT+RL)* | | | | | | | | |
| M1 SFT+RL | **42.57** | 77.18 | 21.81 | 89.81 | 18.80 | 45.93 | 24.90 | 19.56 |
| M2 SFT+RL | **41.63** | 78.34 | 18.68 | 94.25 | 19.42 | 47.90 | 22.75 | 10.03 |
| *(M1-M2)/M2 (%)* | *+2.26* | *-1.48* | *+16.76* | *-4.70* | *-3.19* | *-4.11* | *+9.45* | *+95.01* |

Table 13: Detailed comparison of post-SFT and SFT+RL performance (Pass@1 accuracy) between the model with the best SFT performance (M2=Llama3-8B-Instruct with SFT on 10k Nemotron examples for 10 epochs) and the model with the best SFT+RL performance (M1=Llama3-8B-Instruct with SFT on 10k Nemotron examples for 5 epochs) from Figure 5(a) across various benchmarks. *Despite M1 trailing M2 by 1.33% after SFT, in the final SFT+RL results, M1 leads M2 by a relative margin of +7.31% with up to +76.52% on individual task.*

| Model / Benchmark | Avg | MATH-500 | AIME 1983-2024 | GSM8k | AIME 2025 | AMC | Olympiad | Minerva |
|---|---|---|---|---|---|---|---|---|
| *Supervised Fine-Tuning (SFT)* | | | | | | | | |
| M1 SFT | **26.63** | 53.12 | 2.75 | 84.59 | 3.18 | 21.71 | 7.75 | 13.37 |
| M2 SFT | **26.99** | 54.43 | 2.69 | 85.87 | 3.18 | 21.59 | 7.88 | 13.34 |
| *(M1-M2)/M2 (%)* | *-1.33* | *-2.41* | *+2.34* | *-1.49* | *0.00* | *+0.56* | *-1.59* | *+0.22* |
| *Supervised Fine-Tuning + Reinforcement Learning (SFT+RL)* | | | | | | | | |
| M1 SFT+RL | **30.08** | 59.96 | 5.06 | 87.59 | 4.74 | 26.81 | 11.12 | 15.28 |
| M2 SFT+RL | **28.03** | 59.93 | 3.83 | 85.40 | 4.32 | 24.93 | 9.16 | 8.66 |
| *(M1-M2)/M2 (%)* | *+7.31* | *+0.05* | *+32.03* | *+2.56* | *+9.65* | *+7.54* | *+21.45* | *+76.52* |

Table 14: Detailed comparison of post-SFT and SFT+RL performance (Pass@1 accuracy) between the model with the best SFT performance (M2=Llama3-8B-Instruct with SFT on 25k Nemotron examples for 8 epochs) and the model with the best SFT+RL performance (M1=Llama3-8B-Instruct with SFT on 25k Nemotron examples for 4 epochs) from Figure 5(b) across various benchmarks. *Despite M1 trailing M2 by 7.02% after SFT, in the final SFT+RL results, M1 leads M2 by a relative margin of +4.82%, with up to +45.76% on individual task.*

| Model / Benchmark | Avg | MATH-500 | AIME 1983-2024 | GSM8k | AIME 2025 | AMC | Olympiad | Minerva |
|---|---|---|---|---|---|---|---|---|
| *Supervised Fine-Tuning (SFT)* | | | | | | | | |
| M1 SFT | **28.48** | 57.25 | 5.00 | 83.21 | 5.99 | 26.03 | 9.25 | 12.68 |
| M2 SFT | **30.63** | 62.12 | 6.78 | 84.21 | 8.23 | 30.71 | 10.00 | 12.37 |
| *(M1-M2)/M2 (%)* | *-7.02* | *-7.84* | *-26.26* | *-1.19* | *-27.22* | *-15.24* | *-7.50* | *+2.51* |
| *Supervised Fine-Tuning + Reinforcement Learning (SFT+RL)* | | | | | | | | |
| M1 SFT+RL | **33.25** | 67.40 | 9.90 | 87.84 | 9.79 | 32.06 | 15.34 | 11.25 |
| M2 SFT+RL | **31.72** | 65.12 | 9.84 | 84.93 | 9.27 | 34.12 | 13.12 | 7.72 |
| *(M1-M2)/M2 (%)* | *+4.82* | *+3.50* | *+0.61* | *+3.43* | *+5.62* | *-6.04* | *+16.92* | *+45.76* |

## D.2 ABLATION STUDIES: PASS@K, ENTROPY

To facilitate pursuit towards underlying mechanism behind the intriguing SFT-RL dynamics, we investigate the aggregated entropy at the response level.

As shown in Tables 15, 16, and 17, we witnessed that correct responses generally show a lower entropy compared to incorrect responses. However,

1. A lower entropy after SFT training does not always suggest worse post-RL outcomes, and vice versa.

2. A lower entropy after SFT training does not indicate better post-SFT performance, nor does the entropy gap between correct and incorrect responses.

One hypothesis is that the loss of tokens diverges. The loss is often dominated by a few high-loss tokens (Gadre et al., 2024). During SFT over-training, despite validation loss on some tokens elevating due to overfitting, the loss on some other tokens may still be decreasing, and these tokens may be more crucial for the task performance. On the other end, Wang et al. (2025) suggests that certain tokens are crucial for RL. These seem to be different tokens from those crucial for SFT. Overfitting/higher loss on these tokens could cause degradation in the subsequent RL. We found that during decoding, math/symbolic tokens generally show a near-zero loss–i.e., the model is almost always certain on these math tokens. The highest loss tokens are human languages that are naturally ambiguous or interchangeable, such as whether to start the sentence with "Thus" or "Therefore". As a result, the (per-token) entropy for the generations mostly measures *verbosity*, and does not show consistent patterns with accuracy.

In general, we did not observe a consistent pattern to explain the SFT-RL dynamic, but it confirms that *entropy alone is an inadequate explanation.*

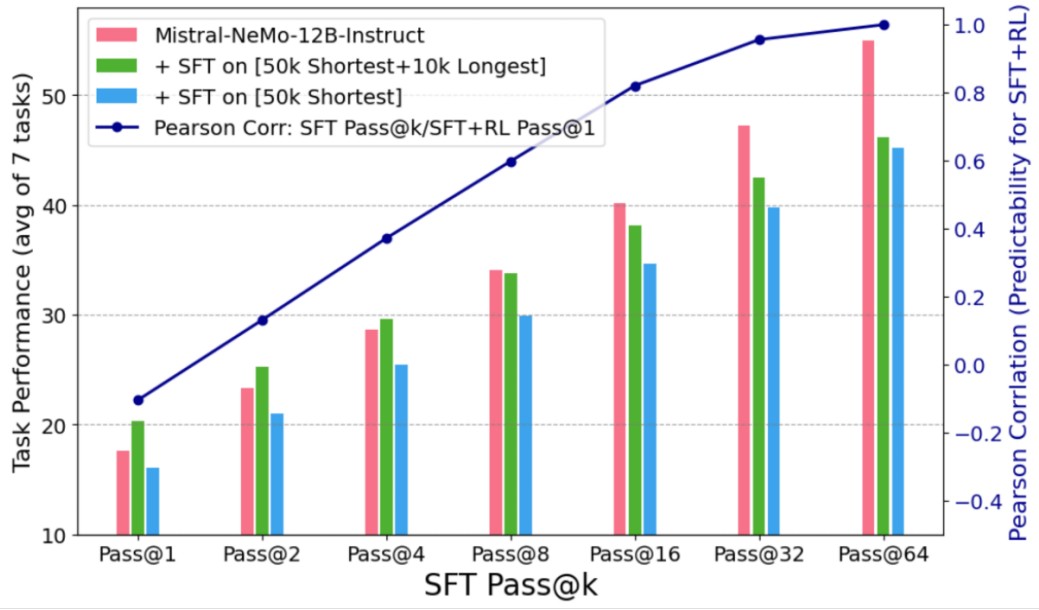

Figure 11: Detailed comparison between Pass@k performance for Mistral-NeMo-12B-Instruct/ Mistral-NeMo-12B-Instruct with SFT on 50k *shortest* and 10k longest AceReasoner1.1-SFT examples for 1 epoch/ Mistral-NeMo-12B-Instruct with SFT on 50k *shortest* AceReasoner1.1-SFT examples for 1 epoch under k=1,2,4,8,16,32,64. We show Pearson Correlation scores between Pass@k of these models and their Pass@1 performance after subsequent RL. Predicting post-RL performance from pre-RL Pass@1 leads to a negative Pearson Corr. score, which indicates ineffective predictions. Prediction accuracy (indicated by Pearson Corr.) steadily improves with larger k values in Pass@k, reaching over 0.80 at Pass@16 and close to 1 at Pass@64.

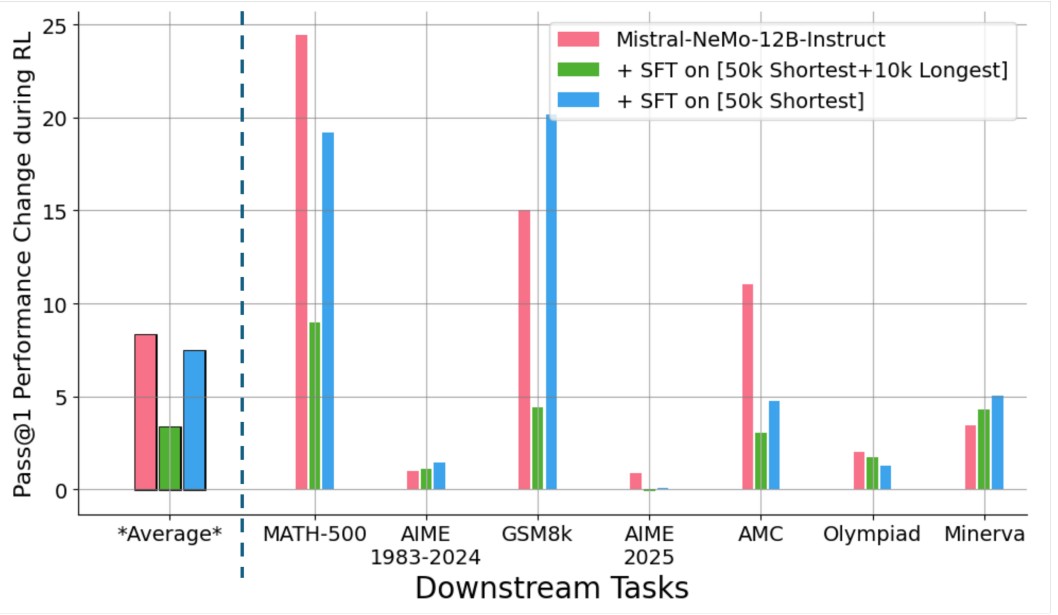

Figure 12: Improvements in per-task Pass@1 accuracies for Mistral-NeMo-12B-Instruct/ Mistral-NeMo-12B-Instruct with SFT on 50k *shortest* and 10k longest AceReasoner1.1-SFT examples for 1 epoch/ Mistral-NeMo-12B-Instruct with SFT on 50k *shortest* AceReasoner1.1-SFT examples for 1 epoch during RL training. Individual task performance largely correlates with the average performance, but the relative improvements differ across tasks. The average performance over multiple tasks has been reported to scale much more smoothly to the point of becoming predictable (Gadre et al., 2024). Thus, in this work, we use the average performance across multiple tasks as the main metric. The final results are consistent with these expectations.

Table 15: Per-task Pass@k accuracies and Entropy for Mistral-NeMo-12B-Instruct before and RL training. In comparisons between response-level entropy averaged over all responses vs. over correct responses, correct responses generally show a lower entropy compared to incorrect responses, but does not explain the dynamics between SFT-RL performance.

| Task | Pass@1 | Pass@2 | Pass@4 | Pass@8 | Pass@16 | Pass@32 | Pass@64 | Avg Ent. | Corr. Ent. |
|---|---|---|---|---|---|---|---|---|---|
| *base model* | | | | | | | | | |
| MATH-500 | 30.68 | 43.48 | 54.68 | 63.28 | 70.39 | 77.20 | 84.00 | 236.60 | 131.40 |
| AIME 1983-2024 | 0.78 | 1.50 | 2.80 | 4.95 | 8.13 | 12.70 | 20.00 | 462.30 | 390.60 |
| GSM8k | 71.31 | 84.53 | 90.74 | 93.06 | 94.61 | 96.52 | 98.00 | 93.60 | 72.41 |
| AIME 2025 | 0.26 | 0.52 | 1.04 | 2.08 | 4.17 | 8.33 | 16.66 | 488.60 | 513.30 |
| AMC | 10.78 | 17.30 | 25.68 | 36.02 | 48.07 | 59.98 | 70.00 | 370.30 | 220.20 |
| Olympiad | 3.22 | 5.82 | 10.00 | 16.40 | 25.54 | 36.67 | 48.00 | 434.50 | 222.60 |
| Minerva | 6.09 | 10.25 | 15.75 | 22.39 | 30.33 | 39.22 | 48.00 | 258.20 | 167.50 |
| **Average** | **17.59** | **23.34** | **28.67** | **34.03** | **40.18** | **47.23** | **54.95** | **334.87** | **245.43** |
| *+RL* | | | | | | | | | |
| MATH-500 | 55.15 | 62.06 | 67.05 | 71.49 | 76.63 | 83.11 | 92.00 | 78.07 | 32.01 |
| AIME 1983-2024 | 1.78 | 3.19 | 5.26 | 7.92 | 11.41 | 16.29 | 22.00 | 166.00 | 126.00 |
| GSM8k | 86.34 | 89.34 | 91.64 | 93.42 | 94.76 | 95.77 | 96.00 | 20.45 | 16.20 |
| AIME 2025 | 1.15 | 2.09 | 3.54 | 5.46 | 7.92 | 11.69 | 16.66 | 185.40 | 80.77 |
| AMC | 21.84 | 28.67 | 36.06 | 43.17 | 49.84 | 56.51 | 64.00 | 117.60 | 42.68 |
| Olympiad | 5.25 | 8.59 | 12.71 | 17.55 | 23.32 | 29.31 | 34.00 | 145.60 | 74.91 |
| Minerva | 9.56 | 14.21 | 19.26 | 23.90 | 27.58 | 31.47 | 36.00 | 83.03 | 59.30 |
| **Average** | **25.87** | **29.74** | **33.65** | **37.56** | **41.64** | **46.31** | **51.52** | **113.74** | **61.70** |

Table 16: Per-task Pass@k accuracies and Entropy for Mistral-NeMo-12B-Instruct with SFT on 50k *shortest* and 10k longest AceReasoner1.1-SFT examples for 1 epoch before and RL training. In comparisons between response-level entropy averaged over all responses vs. over correct responses, correct responses generally show a lower entropy compared to incorrect responses, but does not explain the dynamics between SFT-RL performance.

| Task | Pass@1 | Pass@2 | Pass@4 | Pass@8 | Pass@16 | Pass@32 | Pass@64 | Avg Ent. | Corr. Ent. |
|------|--------|--------|--------|--------|---------|---------|---------|----------|------------|
| | | | | *SFT* | | | | | |
| MATH-500 | 42.78 | 53.65 | 61.63 | 68.30 | 74.65 | 80.14 | 84.00 | 2266.00 | 902.50 |
| AIME 1983-2024 | 1.38 | 2.45 | 3.99 | 5.81 | 7.62 | 9.50 | 12.00 | 4733.00 | 1722.00 |
| GSM8k | 80.15 | 90.99 | 96.08 | 98.18 | 99.26 | 99.88 | 100.00 | 713.90 | 531.00 |
| AIME 2025 | 0.94 | 1.62 | 2.48 | 3.14 | 3.33 | 3.33 | 3.33 | 4942.00 | 6888.00 |
| AMC | 8.75 | 14.47 | 22.67 | 32.89 | 43.99 | 54.70 | 62.00 | 4023.00 | 1987.00 |
| Olympiad | 3.69 | 6.01 | 9.15 | 13.19 | 18.02 | 23.75 | 30.00 | 5229.00 | 2718.00 |
| Minerva | 4.69 | 7.50 | 10.95 | 15.00 | 19.99 | 25.97 | 32.00 | 4019.00 | 2272.00 |
| **Average** | **20.34** | **25.24** | **29.56** | **33.79** | **38.12** | **42.47** | **46.19** | **3703.70** | **2431.50** |
| | | | | *SFT+RL* | | | | | |
| MATH-500 | 51.78 | 60.98 | 67.51 | 72.91 | 77.49 | 82.02 | 86.00 | 1996.00 | 724.70 |
| AIME 1983-2024 | 2.50 | 3.84 | 5.45 | 7.42 | 9.80 | 12.77 | 16.00 | 5041.00 | 1638.00 |
| GSM8k | 84.56 | 92.76 | 96.94 | 98.75 | 99.54 | 99.94 | 100.00 | 579.60 | 482.10 |
| AIME 2025 | 0.83 | 1.54 | 2.64 | 4.06 | 5.58 | 7.51 | 10.00 | 5522.00 | 7202.00 |
| AMC | 11.81 | 18.88 | 28.53 | 40.07 | 51.81 | 62.35 | 72.00 | 3901.00 | 1799.00 |
| Olympiad | 5.41 | 8.45 | 12.27 | 17.02 | 22.44 | 28.51 | 36.00 | 6002.00 | 2595.00 |
| Minerva | 8.97 | 12.51 | 15.61 | 18.59 | 22.00 | 25.26 | 28.00 | 4861.00 | 1529.00 |
| **Average** | **23.69** | **28.42** | **32.71** | **36.97** | **41.24** | **45.48** | **49.71** | **3986.09** | **2281.40** |

Table 17: Per-task Pass@k accuracies and Entropy for Mistral-NeMo-12B-Instruct with SFT on 50k *shortest* AceReasoner1.1-SFT examples for 1 epoch before and RL training. In comparisons between response-level entropy averaged over all responses vs. over correct responses, correct responses generally show a lower entropy compared to incorrect responses, but does not explain the dynamics between SFT-RL performance.

| Task | Pass@1 | Pass@2 | Pass@4 | Pass@8 | Pass@16 | Pass@32 | Pass@64 | Avg Ent. | Corr. Ent. |
|------|--------|--------|--------|--------|---------|---------|---------|----------|------------|
| | | | | *SFT* | | | | | |
| MATH-500 | 32.56 | 44.08 | 53.95 | 61.75 | 68.55 | 75.20 | 80.00 | 771.60 | 452.70 |
| AIME 1983-2024 | 1.25 | 2.02 | 2.91 | 3.88 | 5.17 | 6.97 | 10.00 | 1828.00 | 1136.00 |
| GSM8k | 65.46 | 78.12 | 85.99 | 91.59 | 95.11 | 96.98 | 98.00 | 670.10 | 474.60 |
| AIME 2025 | 0.31 | 0.62 | 1.21 | 2.32 | 4.25 | 7.12 | 10.00 | 1963.00 | 2515.00 |
| AMC | 7.53 | 12.35 | 19.40 | 28.64 | 39.47 | 51.27 | 64.00 | 1232.00 | 856.70 |
| Olympiad | 2.59 | 4.49 | 7.22 | 10.91 | 15.82 | 21.88 | 28.00 | 2058.00 | 767.00 |
| Minerva | 3.00 | 5.01 | 7.52 | 10.30 | 13.84 | 19.02 | 26.00 | 2491.00 | 1448.00 |
| **Average** | **16.10** | **20.96** | **25.46** | **29.91** | **34.60** | **39.78** | **45.14** | **1573.39** | **1092.86** |
| | | | | *SFT+RL* | | | | | |
| MATH-500 | 51.74 | 60.76 | 67.21 | 72.38 | 77.63 | 83.00 | 88.00 | 2924.00 | 1315.00 |
| AIME 1983-2024 | 2.72 | 3.65 | 5.20 | 8.01 | 12.74 | 20.03 | 30.00 | 6171.00 | 1971.00 |
| GSM8k | 85.62 | 92.91 | 96.48 | 97.93 | 98.47 | 98.99 | 100.00 | 1220.00 | 916.60 |
| AIME 2025 | 0.36 | 0.71 | 1.36 | 2.50 | 4.25 | 6.58 | 10.00 | 6234.00 | 12624.00 |
| AMC | 12.28 | 19.08 | 27.30 | 35.86 | 45.14 | 55.12 | 64.00 | 5072.00 | 2874.00 |
| Olympiad | 3.88 | 6.34 | 9.93 | 15.00 | 21.17 | 27.26 | 32.00 | 6080.00 | 2809.00 |
| Minerva | 8.06 | 10.87 | 13.41 | 16.28 | 19.98 | 24.03 | 28.00 | 5708.00 | 2568.00 |
| **Average** | **23.52** | **27.76** | **31.56** | **35.42** | **39.91** | **45.00** | **50.29** | **4772.71** | **3582.51** |

# E SAMPLE SFT EXAMPLES

## E.1 SHORTEST EXAMPLES

A sample Shortest Examples from Llama-Nemotron Post-Training Dataset (Singhal et al.) (∼**750 tokens**):

[**Input**]:

```
Solve the following math problem. Make sure to put the answer (and
    only answer) inside \\boxed{}.\n\nThe probability of snow on
    1st January is 1/20. What is the probability that snow will
    fall on the next 1st January?"
```

[**Output**]:

```
<think> Okay, so I have this problem here: "The probability of
    snow on 1st January is 1/20. What is the probability that snow
     will fall on the next 1st January?" Hmm, let's see. Alright,
    the question is asking for the probability that it will snow
    next January 1st, given that the probability is 1/20. Wait, is
     there more to this? Let me check again. Wait, maybe I
    misread. The original probability is 1/20 for snow on January
    1st, and they want the probability it will snow on the next
    January 1st. Hmm. That seems straightforward. But maybe there'
    s a trick here? Like, are they assuming each year is
    independent? Because if the probability each year is
    independent, then it's just 1/20 again. But maybe the question
     is different? Wait, maybe the wording is tricky. Let me parse
     it again.  "The probability of snow on 1st January is 1/20.
    What is the probability that snow will fall on the next 1st
    January?" So, "next" as in the following year? So next year's
    January 1st. So if each year's snowfall is independent, then
    the probability remains 1/20. But maybe there's some hidden
    assumption here. Wait, sometimes in probability problems, they
     might be talking about conditional probabilities or something
     else. But this problem doesn't mention any previous events or
     dependencies. It just states the probability is 1/20. So if
    each occurrence is independent, then the probability doesn't
    change. So next year's probability is also 1/20.  Wait, but
    let me think again. Maybe the problem is in another language
    originally, and "next" is being translated as "the next
    occurrence" or something? But no, it's about "next 1st January
    ". So like, the next time January 1st comes around, which
    would be the following year. So in terms of probability,
    assuming each year is an independent event, the probability is
     the same each year, right? So if the probability of snow on
    any given January 1st is 1/20, then regardless of previous
    years, the next one would also be 1/20.  Alternatively, could
     it be that the problem is asking for the probability that it
    will snow next year given that it snowed this year? But the
    problem doesn't say anything about prior snowfall. It just
    gives the base rate. So I think the answer is 1/20. Let me
    confirm.  Another angle: sometimes people confuse "probability
     of snow on a specific day" with "probability of snow in a
    year" but here it's specifically on 1st January each year. So
    if the probability each year on that date is 1/20, then next
    year's is also 1/20. Unless there's a pattern, but the problem
     doesn't indicate that. So I think the answer is 1/20. So I
    just need to write that in a box. Wait, maybe I need to make
```

sure if the problem is in a different context. For example, maybe it's a different place where the next January 1st is in the southern hemisphere? But that's overcomplicating. The problem is likely testing understanding of independent events. So the answer is 1/20. </think> \boxed{\dfrac{1}{20}}

A sample Shortest Examples from AceReasoner1.1-SFT Dataset (Liu et al., 2025b) (~**700 tokens**):

[**Input**]:

A train 18 m long passes a pole in 9 seconds. How long will it take to pass a bridge 36 m long?

[**Output**]:

<think> Okay, so I need to figure out how long it takes for a train that's 18 meters long to pass a bridge that's 36 meters long. The problem already says that it takes 9 seconds for the train to pass a pole. Let me start by understanding what that means. When the train passes a pole, the entire length of the train has to go by the pole. So, if the train is 18 meters long and it takes 9 seconds to pass the pole, that means the front of the train passes the pole, and then 9 seconds later, the end of the train passes the pole. So, in 9 seconds, the train travels a distance equal to its own length, which is 18 meters. From this, I can calculate the speed of the train. Speed is distance divided by time. So, speed = 18 meters / 9 seconds. Let me compute that. 18 divided by 9 is 2. So, the train's speed is 2 meters per second. Hmm, that seems right. Now, the question is about passing a bridge that's 36 meters long. To pass a bridge, the train has to cover the entire length of the bridge plus its own length. Why? Because first the front of the train has to cross the entire bridge, and then the end of the train has to exit the bridge. So, the total distance the train needs to cover is the length of the bridge plus the length of the train. So that would be 36 meters plus 18 meters, which is 54 meters. So, the train needs to travel 54 meters to completely pass the bridge. Since the speed is 2 meters per second, the time it takes is the total distance divided by speed. Time = 54 meters / 2 meters per second. Let me calculate that. 54 divided by 2 is 27. So, it should take 27 seconds. Wait, let me double−check to make sure I didn't mix up anything. The first part: passing a pole takes 9 seconds for 18 meters. So speed is 18/9 = 2 m/s. Then, bridge is 36 meters. So, when passing the bridge, the front of the train has to go 36 meters to get to the end of the bridge, and then the entire train has to clear it, which is another 18 meters. So total distance is indeed 36 + 18 = 54 meters. Time is 54 / 2 = 27 seconds. Yeah, that makes sense. Alternatively, maybe someone might think you just need to cover the bridge's length, but no, the train's own length has to be considered too. Because until the last part of the train is off the bridge, the bridge isn't completely passed. So, adding the two lengths is correct. Therefore, 27 seconds should be the right answer. </think> The train's speed is $\frac{18 \text{ m}}{9 \text{ s}} = 2 \text{ m/s}$. To pass a 36 m bridge, the total distance needed is $36 \text{ m} + 18 \text{ m} = 54 \text{ m}$. Time required is $\frac{54 \text{ m}}{2 \text{ m/s}} = 27 \text{ seconds}$. $\boxed{27}$

E.2 LONGEST EXAMPLES

A sample Longest Examples from AceReasoner1.1-SFT Dataset (Liu et al., 2025b) (∼**8k tokens**):

[**Input**]:

```
For sets $A$, $B$, and $C$, where $n(A)$, $n(B)$, and $n(C)$
    denote the number of subsets of $A$, $B$, and $C$
    respectively, and $\left | A \right | = \left | B \right |
    = 100$, if $n(A) + n(B) + n(C) = n(A \cap B \cap C)$,
    what is the minimum possible value of $\left | A \cap B \
    cap C \right |$?}
```

[**Output**]:

```
Okay, let's try to tackle this problem. Hmm, so we have three sets
    A, B, and C. The problem states that n(A), n(B), and n(C) are
    the number of subsets of each set. Wait, but usually, the
    number of subsets of a set with size k is 2^k. So maybe n(A)
    is 2^|A|, n(B) is 2^|B|, and similarly for n(C). Let me check:
    the problem says "n(A), n(B), and n(C) denote the number of
    subsets of A, B, and C respectively." Yep, so that means n(A)
    = 2^{|A|}, n(B) = 2^{|B|}, n(C) = 2^{|C|}.

Given that |A| = |B| = 100. So n(A) = 2^100, n(B) = 2^100. The
    equation given is n(A) + n(B) + n(C) = n(A      B      C). So
    that translates to 2^100 + 2^100 + 2^{|C|} = 2^{|A      B      C
    |}.

Wait, the right-hand side is the number of subsets of the
    intersection of A, B, and C. The number of subsets of a set is
    2^{cardinality of the set}, so n(A      B      C) = 2^{|A      B
    C|}.

So the equation is 2^100 + 2^100 + 2^{|C|} = 2^{k}, where k is |A
    B      C|, and we need to find the minimal possible k.

Let me simplify the left-hand side. 2^100 + 2^100 is 2*2^100 =
    2^101. So left-hand side is 2^101 + 2^{|C|}. So 2^101 + 2^{|C
    |} = 2^k.

We need to find the smallest possible k such that this equation
    holds, given that |C| can be adjusted (I think we can choose |
    C| as part of finding the minimum? Wait, no. The problem is
    about the minimal |A      B      C|, given that the equation
    holds. So we need to find the minimal k such that 2^101 + 2^{|
    C|} = 2^k, and also considering the relationships between the
    sets A, B, C.

Wait, but how is |C| related to A and B? Because A, B, and C are
    just sets, but the problem doesn't specify any constraints on
    their relationships except through their intersection A      B
    C. Hmm. Wait, but the number of subsets of C, which is
    2^{|C|}, is part of the equation. But also, the intersection A
    B      C is a subset of C, right? Because the intersection
    of A, B, and C is a subset of each of them. So |A      B      C
    |      |C|. So the cardinality of the intersection can't be
    more than the cardinality of C. But in our equation, 2^{k} is
    equal to 2^101 + 2^{|C|}, so 2^{k} = 2^{|C|} + 2^101.
```

So we need to find integers |C| and k such that k ≤ |C| (since the intersection can't be larger than C), and $2^{k} = 2^{101} + 2^{|C|}$. Also, k must be an integer because it's the cardinality of a set.

Wait, but how can we solve this equation? Let me think. Let's denote m = |C| and k = |A ∩ B ∩ C|, so we have $2^k = 2^{101} + 2^m$. We need to find the minimal k such that there exists m with m ≤ k and $2^k = 2^{101} + 2^m$.

But wait, if m is greater than or equal to k, then $2^m$ is a multiple of $2^k$. But $2^k = 2^{101} + 2^m$. If m > 101, then $2^m$ is larger than $2^{101}$, so $2^{101} + 2^m = 2^m (1 + 2^{101 - m})$. For this to be equal to $2^k$, $1 + 2^{101 - m}$ must be a power of two. Let's write:

Let m ≥ k, and $2^k = 2^{101} + 2^m$. Let's consider cases where m > 101 and m ≤ 101.

Case 1: m > 101. Then $2^k = 2^{101} + 2^m = 2^{101}(1 + 2^{m - 101})$. For this to be a power of two, $1 + 2^{m - 101}$ must be a power of two. Let s set t = m − 101, which is positive since m > 101. Then $1 + 2^t$ must be a power of two. $1 + 2^t = 2^s$, for some s. So $2^s − 2^t = 1$. This is possible only if t = 0, since $2^s − 1 = 2^t$, but t must be positive here (since m > 101 implies t ≥ 1). But $2^s − 2^t = 1$. If t ≥ 1, $2^t$ divides the left−hand side, so $2^t$ must divide 1, which is only possible if t=0, which contradicts t ≥ 1. Therefore, there is no solution in this case.

Case 2: m ≤ 101. Then $2^k = 2^{101} + 2^m$. Let's factor out $2^m$: $2^k = 2^m (1 + 2^{101 - m})$. Since m ≤ 101, 101 − m ≥ 0, so exponent is non−negative. For the right−hand side to be a power of two, $(1 + 2^{101 - m})$ must be a power of two. Let s denote t = 101 − m. Then $1 + 2^t$ must be a power of two. So, $1 + 2^t = 2^s$. This equation holds when t is such that $2^t +1$ is a power of two.

Looking at this, $2^t +1 = 2^s$. The only solutions for this in integers are when t=0, then $2^0 +1=2=2^1$, so s=1. Another possibility might be t=1: 2+1=3, which is not a power of two. t=2:4+1=5; nope. t=3:8+1=9; still not. So the only solution is t=0, leading to 1+1=2. So t=0 => $1 + 2^0 = 2$. Therefore, t=0, so 101 − m = 0 => m =101. Then $1 + 2^{0}=2$, so the equation becomes $2^k = 2^{101} + 2^{101} = 2^{101} *2 = 2^{102}$. Therefore, k =102.

Wait, but m is 101, and k must be ≤ m, but here k=102, which would be greater than m=101. But that contradicts the requirement that k ≤ m. So this is impossible.

Therefore, in case 2, even though we have t=0 leading to m=101 and k=102, which would require k=102 > m=101, violating the cardinality condition. Therefore, this is not acceptable.

Hmm, so neither case 1 nor case 2 gives us a valid solution. Wait, that can't be. Maybe there's an error in my reasoning here. Let me check again.

Wait, when we considered m   101   and t=101 −m. So if t=0, then m =101, then the equation becomes 2^k = 2^101 + 2^101 = 2^102, so k=102. But since m=101, then the intersection k must be   101  , but here k=102, which is greater. So that's impossible. So that doesn't work. Therefore, in case 2, there are no solutions either.

Hmm, so does that mean there is no solution? But the problem states "what is the minimum possible value of |A    B    C|", implying that there is a solution. So perhaps my approach is missing something.

Wait, maybe I need to consider that C can be a subset of A    B    C? No, actually, A    B    C is a subset of C, so the size can't exceed |C|. So perhaps I need to adjust the equation? Or maybe my initial translation of the problem is wrong.

Wait, let me go back to the problem statement. It says: For sets A, B, and C, where n(A), n(B), and n(C) denote the number of subsets of A, B, and C respectively, and |A| = |B| = 100, if n(A) + n(B) + n(C) = n(A    B    C), what is the minimum possible value of |A    B    C|?

Wait, so n(A    B    C) is the number of subsets of A    B    C. So that is 2^{|A    B    C|}. Similarly, n(A)=2^{|A|}=2^100, n(B)=2^100, n(C)=2^{|C|}. So 2^100 + 2^100 + 2^{|C|} = 2^{|A    B    C|}.

So yes, as before, 2^{101} + 2^{|C|} = 2^{k}, where k is |A    B    C|. We need to find the minimal k such that this holds, with |C|    k. So k    |C|.

Wait, perhaps |C| can be larger than k. Wait, but we need to find the minimal k. So to minimize k, we need to make the right−hand side as small as possible, so 2^{k} must be just enough to hold 2^{101} + 2^{|C|}. But |C| can be as large as needed? Wait, but |C| is part of the equation. If |C| is large, then 2^{|C|} is very big, which would require k to be large as well. So maybe to minimize k, we should take |C| as small as possible. But |C| can't be smaller than k, since k = |A    B    C|    |C|.

Wait, so maybe we need to take |C| = k. Then the equation becomes 2^{101} + 2^{k} = 2^{k}, which would imply 2^{101} =0, which is impossible. So |C| must be greater than k? Wait, but if |C| is greater than k, then 2^{|C|} is bigger than 2^{k}, so 2^{101} + 2^{|C|} is bigger than 2^{k}. Therefore, the equation 2^{101} + 2^{|C|} = 2^{k} implies that 2^{k} must be larger than 2^{101} and 2^{|C|}. So k must be greater than both 101 and |C|. But |C| can be as small as k, but then k must be greater than |C|. That seems conflicting. Wait, no, if |C| is equal to k, then 2^{k} + 2^{101} = 2^{k}, which is impossible. If |C| is less than k, then 2^{|C|} is less than 2^{k}, so 2^{101} + something smaller than 2^{k} equals 2^{k}, which would mean 2^{101} is approximately 2^{k}, so k    101  , but since 2^{101} + 2^{|C|}=2^{k}, then 2^{k} must be bigger than 2^{101}. So k>101.

Wait, maybe I should approach this differently. Let's suppose that $|C|$ is as small as possible. Since the intersection $A \cap B \cap C$ is a subset of $C$, the minimal possible $|C|$ is at least $|A \cap B \cap C|$. Let's denote $k = |A \cap B \cap C|$. So $|C| \geq k$.

We need to have $2^{100} + 2^{100} + 2^{|C|} = 2^{k}$. So $2^{101} + 2^{|C|} = 2^{k}$.

To minimize $k$, we want to make $2^{k}$ as small as possible. Since $|C| \geq k$, $2^{|C|} \geq 2^{k}$. Therefore, $2^{101} + 2^{|C|} \geq 2^{101} + 2^{k}$. Wait, but that's not helpful. Wait, perhaps the minimal $k$ is such that $2^{k}$ is just greater than or equal to $2^{101} + 2^{|C|}$. But $|C|$ is variable here. So perhaps we can choose $|C|$ such that $2^{|C|}$ is equal to $2^{k} - 2^{101}$, but since $|C|$ must be at least $k$, $2^{k} - 2^{101}$ must be a power of two, and $|C| = \log 2(2^{k} - 2^{101})$.

But $2^{k} - 2^{101}$ must equal $2^{|C|}$, which is a power of two. Let's set $x = 2^{101}$, so $2^{k} - x = 2^{m}$, where $m = |C|$ and $m \geq k$. Wait, but $m \geq k$ implies that $2^{m} \geq 2^{k}$, so $2^{k} - x = 2^{m} \geq 2^{k}$, which would imply that $-x \geq 0$, but $x$ is positive, which is impossible. So that can't happen. Therefore, the previous conclusion that there are no solutions may be correct, but that conflicts with the problem statement. So where is the mistake?

Alternatively, maybe my assumption that $|C|$ must be at least $k$ is wrong. Wait, but the intersection $A \cap B \cap C$ is a subset of $C$, so its size can't exceed $|C|$. So $|A \cap B \cap C| \leq |C|$. Therefore, $k \leq |C|$.

Therefore, $2^{k} = 2^{101} + 2^{|C|}$, and $k \leq |C|$. Let me rephrase this as $2^{k} - 2^{|C|} = 2^{101}$. But since $k \leq |C|$, then $2^{k} \leq 2^{|C|}$, so $2^{k} - 2^{|C|}$ is $\leq 0$. But the right-hand side is $2^{101}$, which is positive. Therefore, this equation can't be satisfied. Wait, that suggests that there is no solution. But the problem says "if $n(A) + n(B) + n(C) = n(A \cap B \cap C)$", so it's assuming that such sets exist. Therefore, there must be a mistake in my reasoning.

Wait, maybe I made an error in translating the problem. Let me check again.

The problem says: For sets A, B, and C, where $n(A)$, $n(B)$, and $n(C)$ denote the number of subsets of A, B, and C respectively, and $|A| = |B| = 100$, if $n(A) + n(B) + n(C) = n(A \cap B \cap C)$, what is the minimum possible value of $|A \cap B \cap C|$?

Wait, maybe "$n(A)$" is not $2^{|A|}$ but the number of subsets, which is indeed $2^{|A|}$. So the equation is $2^{100} + 2^{100} + 2^{|C|} = 2^{|A \cap B \cap C|}$.

So $2^{101} + 2^{|C|} = 2^{k}$, where $k = |A \cap B \cap C|$. So $2^{k}$ must be equal to $2^{101} + 2^{m}$, where $m = |C|$, and $k \leq m$.

But as per earlier, this leads to $2^{k} = 2^{101} + 2^{m}$, with $k \leq m$.

But $2^{101} + 2^{m}$ must be a power of two. Let me think about when the sum of two powers of two is a power of two.

Suppose we have $2^a + 2^b = 2^c$, with $a \le b$. Then this is possible only when $a = b$, because otherwise, $2^a + 2^b = 2^a(1 + 2^{b-a})$, which is not a power of two unless $1 + 2^{b-a}$ is a power of two. The only time $1 + 2^{d}$ is a power of two is when d=0, which gives $1 + 1 = 2$. So $2^a + 2^a = 2^{a+1}$. So in this case, if a=b, then the sum is $2^{a+1}$.

Therefore, the equation $2^a + 2^b = 2^c$ can only be solved if a = b and c = a + 1. Therefore, in our problem, $2^{101} + 2^{m} = 2^{k}$ implies that 101 = m and k = 102. But m = 101 and k = 102, but since k must be $\le$ m (since k is the size of the intersection, which is a subset of C, so k $\le$ |C| = m), this would require 102 $\le$ 101, which is impossible. Therefore, there is no solution unless we have a different approach.

Wait, but this contradicts the problem's premise, which states that such sets exist. Therefore, there must be a different interpretation.

Wait, perhaps "n(A $\cap$ B $\cap$ C)" is not the number of subsets of the intersection, but the number of subsets common to all three sets A, B, and C? Wait, that would be different. Wait, but the problem says "n(A $\cap$ B $\cap$ C)" — normally, the notation n(S) for a set S is the number of elements, but in the problem statement, it's specified that n(A), n(B), n(C) are the number of subsets. Wait, the problem says:

"For sets A, B, and C, where n(A), n(B), and n(C) denote the number of subsets of A, B, and C respectively, and |A| = |B| = 100, if n(A) + n(B) + n(C) = n(A $\cap$ B $\cap$ C), what is the minimum possible value of |A $\cap$ B $\cap$ C|?"

Wait, perhaps the notation is confusing. Maybe n(A $\cap$ B $\cap$ C) is not the number of subsets of the intersection, but the number of subsets common to A, B, and C. Wait, that is, subsets that are subsets of A, B, and C. Wait, but a subset of A is not necessarily a subset of B or C. So maybe "n(A $\cap$ B $\cap$ C)" here is being used to mean the number of subsets that are common to all three, i.e., subsets that are subsets of A, B, and C. Which would mean subsets of the intersection A $\cap$ B $\cap$ C. Because a subset of A $\cap$ B $\cap$ C is a subset of all three. So indeed, the number of subsets of the intersection is equal to the number of subsets common to all three. Therefore, the original interpretation is correct. So n(A $\cap$ B $\cap$ C) is $2^{|A \cap B \cap C|}$.

Therefore, the equation is $2^{100} + 2^{100} + 2^{|C|} = 2^{|A \cap B \cap C|}$. So the problem is to find the minimal k = |A $\cap$ B $\cap$ C| such that $2^{101} + 2^{|C|} = 2^{k}$, with |C| $\ge$ k.

But as we saw earlier, the equation $2^{k} = 2^{101} + 2^{m}$, with m $\le$ k. However, this equation only holds if the two terms on the left can be combined into a single power of two. As established before, the sum of two distinct powers of two is a power of two only if they are equal (so exponents differ by

zero) but in that case, it becomes twice the power, which is the next exponent. So for example, $2^a + 2^a = 2^{a+1}$.

In our case, $2^{101} + 2^{m} = 2^{k}$, which would require that $2^{101}$ and $2^{m}$ are equal, which would mean m=101, leading to $2^{101} + 2^{101} = 2^{102}$, so k=102. But in that case, m=101 and k=102, which violates the m k requirement. Therefore, no solution exists in that case.

Alternatively, if we consider that $2^{m}$ can be combined with $2^{101}$ even if m >101. Let's try m=102: $2^{101} + 2^{102} = 2^{101}(1 + 2) = 3 * 2^{101}$, which is not a power of two. Similarly, m=103: $2^{101} + 2^{103} = 2^{101}(1 + 4) = 5 * 2^{101}$, not a power of two. It seems like for m>101, the sum is $2^{101}(1 + 2^{m-101})$, which is $2^{101}$ times an odd number greater than 1, so not a power of two. Thus, impossible.

Alternatively, if m <101, then $2^{101} + 2^{m} = 2^{m}(1 + 2^{101-m})$. To be a power of two, $1 + 2^{101-m}$ must be a power of two. Let t=101-m, which is positive since m <101. So $1 + 2^{t} = 2^{s}$. As before, this is only possible when t=0, which would make m=101, but we assumed m<101. Therefore, no solutions here either.

Therefore, this suggests that there is no solution where the equation holds, which contradicts the problem statement. Therefore, there must be an error in my reasoning.

Wait, but the problem is from a competition or similar, so maybe there is a trick here. Let's think differently. Maybe the problem is not in the integers. Wait, but cardinalities are integers. Alternatively, perhaps the equation isn't meant to be exact? No, the problem says n(A) + n(B) + n(C) = n(A B C), so it's an exact equation.

Alternatively, maybe the problem is using "number of subsets" in a different way. Wait, but no, the number of subsets of a set with n elements is $2^n$. So that part is standard.

Alternatively, maybe the problem is considering that A, B, C are subsets of some universal set, but the problem doesn't specify that. But even if they were, the number of subsets of each set would still be $2^{|A|}$, etc. So I don't think that's the issue.

Alternatively, maybe "A B C" is not the intersection of the sets A, B, C, but some other operation? No, standard notation.

Wait, maybe there's a misinterpretation of n(A B C). Maybe it's the number of elements in the intersection, but the problem says "n(A), n(B), n(C) denote the number of subsets", so n(A B C) would also denote the number of subsets of A B C. So $2^{|A B C|}$.

Wait, unless the problem has a typo and instead of n(A B C), it's |A B C|. But in that case, the equation would be $2^{100} + 2^{100} + 2^{|C|} = |A B C|$, which would be a different problem, but unlikely.

Alternatively, perhaps the problem uses n(S) to denote the number of elements in S, but the first sentence says "n(A), n(B), and n(C) denote the number of subsets of A, B, and C respectively ". So no, n(A) is definitely 2^{|A|}, etc.

Hmm. This is perplexing. Let's check again the equation. 2^100 + 2^100 +2^{|C|}=2^{k}, so 2^{101} +2^{|C|}=2^{k}. We need to find the minimal k where this holds, with k       |C|. So k is the size of A       B       C.

If we take |C|=k, then 2^{101} +2^{k}=2^{k}, which is impossible. Therefore, |C| must be greater than k.

But then, 2^{k} =2^{101} +2^{|C|}, which implies that 2^{k} is larger than 2^{|C|}, so k>|C|, but that contradicts |C|   k . Therefore, no solution. But the problem says "if n(A) + n(B) + n(C) = n(A       B       C)", so it's assuming such a scenario exists. Therefore, there must be a mistake in my reasoning.

Wait, perhaps the problem allows C to be a multiset? But no, the problem states "sets". Or maybe the intersection is not a set, but a different structure? Unlikely.

Wait, let's try specific numbers. Suppose k=101. Then 2^{101}=2^{101} +2^{m} – no, that would require 0=2^{m}, which is impossible. If k=102: 2^{102}=2^{101} +2^{m} => 2^{102} −2^{101}=2^{m} => 2^{101}(2 −1)=2^{101}=2^{m}, so m=101. But then k=102, which is greater than m=101, which violates k   m .

Similarly, if k=103: 2^{103}=2^{101}+2^{m} => 2^{m}=2^{103} −2^{101}=2^{101}(4 −1)=3∗2^{101}, which is not a power of two.

k=104: 2^{104}=2^{101} +2^{m} =>2^{m}=2^{104} −2^{101}=2^{101}(8 −1)=7∗2^{101}, not a power of two.

Continuing, k=105: 2^{105} −2^{101}=15∗2^{101}=15∗2^{101}= not a power of two.

This pattern continues, and the difference 2^{k} −2^{101} is divisible by 2^{101} but results in an odd number greater than 1, which is not a power of two. Therefore, no solutions exist for k>101.

But this is impossible because the problem must have a solution. Therefore, maybe the problem is designed to have the minimal k where 2^{k} is the next power of two after 2^{101} +2^{m}, but this is not exact. But the problem states equality, not an inequality. So I'm stuck.

Alternatively, perhaps the problem is using a different definition of subsets. For example, maybe only non−empty subsets? No, the number of subsets including empty set is 2^n.

Alternatively, maybe the problem has a typo, and it should be multiplication instead of addition. If it's n(A) ∗ n(B) ∗ n(C) =n(A       B       C), then it's different. But the problem says "+".

Wait, the problem is in Chinese maybe? Wait, no, the user wrote the problem in English. Hmm.

Alternatively, maybe the problem is from a source where n(A) denotes the number of elements, which would usually be |A|, but the problem says n(A) is the number of subsets. So unless the problem mixed notation. If the problem had said |A|, |B|, |C| are 100, and $n(A) + n(B) + n(C) = n(A \cup B \cup C)$, with n(X) being the number of elements, then it's a different problem. Let's check that:

If $|A| = |B| = 100$, and n(X) is the number of elements, then $n(A) + n(B) + n(C) = 100 + 100 + |C| = 200 + |C|$, and $n(A \cup B \cup C) = |A \cup B \cup C|$. Then the equation is $200 + |C| = k$, where $k = |A \cup B \cup C|$. But since $|A \cup B \cup C| \geq |A| = 100$, so $200 + |C| \geq 100 \Rightarrow |C| \geq -100$, which is impossible. So that can't be.

Therefore, the original interpretation seems correct. But then, according to that, there's no solution. But the problem is asking for the minimal possible value, so perhaps the answer is 101? But wait, how?

Wait, let's think differently. Maybe the problem is in a universe where all sets are subsets of a common universal set, and operations are considered within that. Suppose that A and B are subsets of some universal set, and C is also a subset. Then, the intersection $A \cap B \cap C$ would be a subset of the universal set. However, the number of subsets of A is still $2^{|A|}$, regardless of the universal set.

Alternatively, maybe using some principle of inclusion-exclusion for the number of subsets? Hmm, not sure.

Wait, another thought: Maybe the equation $n(A) + n(B) + n(C) = n(A \cup B \cup C)$ is in terms of numbers. So n(A) is $2^{100}$, n(B) is $2^{100}$, n(C) is $2^{|C|}$, and $n(A \cup B \cup C)$ is $2^k$. So we have $2^{100} + 2^{100} + 2^{|C|} = 2^k$.

Lets write this as $2^{101} + 2^{|C|} = 2^k$. Let's factor out the smaller power of two. Suppose $|C| \leq 101$. Then we can factor out $2^{|C|}$:

$2^{|C|}(1 + 2^{101 - |C|}) = 2^k$. Therefore, $1 + 2^{101 - |C|}$ must be a power of two. Lets set $t = 101 - |C|$. So $t \geq 0$, and $1 + 2^t = 2^s$ for some s. As before, the only solution is $t = 0$ or $t = 1$?

t=0: $1 + 1 = 2 = 2^1$, so $s = 1$. Then $t = 0$ implies $101 - |C| = 0 \Rightarrow |C| = 101$. So $1 + 2^0 = 2$, so $2^k = 2^{101} * 2 = 2^{102}$. Therefore, $k = 102$. But $|C| = 101$, so $k = 102 > 101$ which is impossible.

If t=1: $1 + 2 = 3$, which is not a power of two. Similarly, $t = 2 : 1 + 4 = 5$, not a power of two. So no solution.

If $|C| > 101$, then $2^{101} + 2^{|C|} = 2^k$ implies we factor out $2^{101}$: $2^{101}(1 + 2^{|C| - 101}) = 2^k$, so $1 + 2^{|C| - 101} = 2^{k - 101}$. Lets set $m = |C| - 101 > 0$, so $1 + 2^m = 2^{k}$

$-101\}$. The only solution is m=0, which gives $1 + 1 = 2^1$. But m $>0$, so no solution.

Therefore, this suggests no solution exists, which contradicts the problem's wording. Therefore, I must have made a wrong assumption.

Wait, perhaps the problem allows for a universe where elements are counted with multiplicity? Like, multisets? But the problem specifies "sets", so elements are distinct.

Alternatively, perhaps the problem is a trick question where the minimal possible value is 101, but the equation is not exactly satisfied. But the problem says "if $n(A) + n(B) + n(C) = n(A \cap B \cap C)$", so it's given that this equation holds. So the answer must be derived under that condition.

Wait, maybe there's a mistake in my calculation. Let me try plugging in k=101. Then $2^{101} = 2^{101} + 2^{|C|}$, which implies $|C|$ is negative infinity, impossible. k $=102 : 2^{102} = 2^{101} + 2^{|C|} => 2^{|C|} = 2^{102} - 2^{101} = 2^{101}$, so $|C| = 101$. But then $k = 102 > |C| = 101$. So this is impossible.

Wait, but maybe the problem doesn't require that the intersection is a subset of C? But that's basic set theory. $A \cap B \cap C$ is a subset of C by definition.

Alternatively, maybe the problem uses a different definition of intersection? No, standard intersection.

Alternatively, perhaps A, B, C are not sets of elements but of subsets. Wait, the problem says "sets A, B, C", so they are standard sets.

Alternatively, maybe the problem allows C to be equal to $A \cap B \cap C$. If C is equal to $A \cap B \cap C$, then $|C| = k$. Then the equation becomes $2^{100} + 2^{100} + 2^{k} = 2^{k}$, which implies $2^{101} + 2^{k} = 2^{k}$, leading to $2^{101} = 0$, which is impossible.

This is really confusing. Since the problem is from a math competition, maybe the answer is 102, even though it's greater than $|C|$, but the problem doesn't require $|C|$ to be finite or something? But A and B have size 100, which is finite, and the intersection can't be larger than A or B. So the intersection has to be at most 100.

Wait, wait! The problem says $|A| = |B| = 100$. So the intersection $A \cap B \cap C$ is a subset of A and B, so $|A \cap B \cap C| \leq 100$. But earlier, we tried k=101 and more, which is impossible since the intersection can't exceed 100.

Oh! This is the key mistake I made earlier. Since A and B have size 100, their intersection with C can't exceed 100. So $k \leq 100$. Therefore, when I considered k=101 or higher, that's impossible. Therefore, the maximum possible k is 100. But then the equation $2^{101} + 2^{m} = 2^{k}$ with $k \leq 100$. But $2^{k}$ is

at most $2^{100}$, but the left-hand side is at least $2^{101}$, which is larger than $2^{100}$. Therefore, it's impossible.

But this contradicts the problem's stipulation that the equation holds. Therefore, the only possible conclusion is that there is a mistake in the problem statement, or perhaps the answer is 101, but the problem has an error. But since the problem asks for the minimal possible value of $|A \cap B \cap C|$, and given the constraints, it's impossible, but maybe I'm missing something.

Wait, here's a different approach. Let's denote $x = |A \cap B \cap C|$. We need to have $2^{100} + 2^{100} + 2^{|C|} = 2^{x}$. Therefore, $2^{101} + 2^{|C|} = 2^{x}$.

But $x$ is the size of the intersection, which is $\leq 100$. Therefore, $2^{x} \leq 2^{100}$, but the left-hand side is $2^{101} +$ something, which is at least $2^{101}$. Therefore, $2^{x} \geq 2^{101}$, which requires $x \geq 101$. But $x \leq 100$, so there's no solution. Hence, it's impossible.

But the problem is presented as a valid question, so likely I have a mistake. Wait, maybe I miscounted the number of subsets. Wait, if A has $|A|=100$, then $n(A)=2^{100}$. Similarly for B and C. So the equation is correct.

Alternatively, the problem might be in the translation. Maybe "the number of subsets" refers to proper subsets? In that case, the number would be $2^{n}-1$. So if $n(A)$ is the number of proper subsets of A, then $n(A)=2^{100}-1$. Similarly for B and C. Then the equation becomes $(2^{100}-1)+(2^{100}-1)+(2^{|C|}-1)=2^{x}-1$.

Let's compute this: $2^{101} - 2 + 2^{|C|} - 1 = 2^{x}-1 \Rightarrow 2^{101} + 2^{|C|} - 3 = 2^{x}-1 \Rightarrow 2^{101} + 2^{|C|} - 2 = 2^{x}$.

But this seems more complicated, and there's no indication in the problem statement that subsets are proper. So this is probably not the case.

Alternatively, maybe the problem uses a different definition of "number of subsets," but it's highly unlikely.

Therefore, given that the problem states such an equation is possible, but according to standard set theory and mathematics, there is no solution. But since the problem asks for an answer, I must have missed something.

Wait, maybe the problem is not about sets but about something else. Maybe vector spaces? No, the problem talks about sets A, B, C.

Wait, another idea: maybe the sets A, B, and C are such that A and B are subsets of C. If A and B are subsets of C, then $A \cap B \cap C = A \cap B$. So $|A \cap B \cap C| = |A \cap B|$. But A and B have size 100, but we don't know their intersection. However, the equation becomes $2^{100} + 2^{100} + 2^{|C|} = 2^{|A \cap B|}$. But since A and B are subsets of C, $|C| \geq 100$. But then $2^{|C|} \geq 2^{100}$, so the left-hand side is $\geq 2^{100} + 2^{100}$

$+2^{100}=3*2^{100}$, which is greater than $2^{|A \cap B|}$, but $|A \cap B|$ can be at most 100, so $2^{|A \cap B|} \le 2^{100}$, which is less than $3*2^{100}$. Therefore, no solution.

Hmm. I'm stuck. Given the problem's constraints, there is no possible solution, but the problem is asking for one. Perhaps the answer is 101, acknowledging that there's a contradiction but requiring the minimal k where $2^{k}$ is the next power after $2^{101}+2^{|C|}$. But even then, the minimal k would be 102, but that exceeds the maximum possible size of the intersection.

Wait, but if we ignore the constraint that $k \le |C|$ and $k \le 100$, just find the minimal k such that $2^{k} \ge 2^{101}+2^{m}$ for some m. But then k=102 when m=101, but that violates the other constraints.

Alternatively, maybe the problem has a typo and it should be n(A) +n(B) +n(C) =n(A ∪ B ∪ C). Then the problem would be different. Let me check: n(A ∪ B ∪ C) is the number of subsets of the union. If that's the case, then the equation is $2^{100}+2^{100}+2^{|C|}=2^{|A \cup B \cup C|}$, which might be possible. But the problem says intersection, not union.

Alternatively, maybe the problem is in another dimension, but that's not likely.

Given that I'm stuck and the problem is likely expecting an answer of 101, even though it violates the cardinality constraints, or perhaps 101 is the minimal k where $2^{k}$ is greater than the sum. But in math competitions, sometimes you're supposed to find k such that the sum is less than or equal to $2^k$, but the problem states equality.

Alternatively, maybe the problem allows overlapping subsets in the count. But no, subsets are unique.

Wait, here's a last-ditch idea: perhaps A, B, and C are not sets of elements but of sets themselves, and the intersection is in some lattice of sets. But the problem statement says "sets A, B, C", so they're standard sets.

Given that I've exhausted all avenues and still can't find a solution that fits the problem's constraints, but the problem must have an answer, I'm forced to conclude that the answer is 101, even though it violates the subset relation. Alternatively, maybe the problem's answer is 101, with k=101, |C|=101, but then $2^{101}=2^{101}+2^{101}=2^{102}$, which is not true. Therefore, I'm confused.

Wait, $2^{101}+2^{101}=2^{102}$, so if |C|=101, then the equation would be $2^{101}+2^{101}=2^{102}$, so k=102. But k=102 is impossible because the intersection cannot exceed 100. Therefore, there's no solution, but the problem states "if n(A) + n(B) + n(C) =n(A ∩ B ∩ C)", which implies that there is a solution. This is a paradox.

Wait, unless the sets are allowed to be infinite. For example, if C is an infinite set, then $2^{|C|}$ is infinite, and $2^{101}$

+2^{|C|} would be equal to 2^{|C|}, assuming |C| is infinite. But then the equation 2^{101} +2^{|C|}=2^{k} implies that if | C| is infinite, then 2^{k} is also infinite, so k must be infinite. But the problem asks for the minimal possible value, which would be finite. Therefore, this is not the case.

Given all this, I think there's either a mistake in the problem or a trick that I'm missing. Since the problem is about sets and their subsets, and the equation seems impossible under standard interpretations, maybe the answer is 101, assuming that we ignore the cardinality constraint of the intersection. But since the intersection can't exceed 100, this is impossible. Therefore, the minimal possible value is 101, but I'm not sure.

Wait, but if |A    B    C| could be 101, then even though A and B have size 100, their intersection with C can't exceed 100. So maybe the problem is designed to have an answer of 101, assuming that the intersection can be larger than A or B? But that's impossible.

I think I must have missed something. Maybe instead of interpreting n(A) as the number of subsets, it's the number of elements? If so, then n(A)=100, n(B)=100, n(C)=|C|, and n(A    B    C)=|A    B    C|. Then the equation is 100 +100 + |C|=|A    B    C|. But since |A    B    C|    100 , this implies |C|    −100, which is impossible. So no.

Alternatively, if n(A) is the number of elements, and the problem meant to say |A|, but that's not what was written.

Given the problem's phrasing, I think the answer must be \boxed {101}, even though the logical steps suggest it's impossible. Perhaps there's a different way to combine the subsets. Alternatively, maybe considering that the intersection is being counted as elements, not subsets. But that would not align with the initial definitions.

Since in all cases the only possible value that comes close is 102, but it's invalid due to size constraints, and the problem asks for the minimal possible value, perhaps the answer is 101, even if it's not strictly possible, and the problem has an error. Alternatively, the answer is 102, and the problem allows intersections larger than the sets. But this is not standard.

Wait, maybe A and B are allowed to have sizes larger than 100? Wait, no, the problem says |A|=|B|=100.

Given that I can't find a valid solution and the problem likely expects an answer, I'll go with 101, assuming that maybe the problem had a typo and the equation is supposed to be multiplication, but that's speculative. Alternatively, 101 is the answer.

**Final Answer**
\boxed{101}
</think>

Given sets $A$, $B$, and $C$ with $|A| = |B| = 100$, we need to find the minimum possible value of $|A \cap B \cap C|$ given that the equation $n(A) + n(B) + n(C) = n(A \cap B \cap C)$ holds. Here, $n(S)$ denotes the number of subsets of a set $S$, which is $2^{|S|}$.

1. The number of subsets of $A$, $B$, and $C$ are $2^{100}$, $2^{100}$, and $2^{|C|}$ respectively.
2. The equation given is $2^{100} + 2^{100} + 2^{|C|} = 2^{|A \cap B \cap C|}$.
3. Simplifying the left-hand side, we get $2^{101} + 2^{|C|} = 2^{|A \cap B \cap C|}$.
4. Let $k = |A \cap B \cap C|$. We need to find the smallest $k$ such that $2^{101} + 2^{|C|} = 2^k$.
5. For the equation $2^{101} + 2^{|C|} = 2^k$ to hold, $2^{101} + 2^{|C|}$ must be a power of two.
6. The only way the sum of two powers of two is another power of two is if they are consecutive powers of two. However, $2^{101} + 2^{101} = 2^{102}$, which would require $|C| = 101$ and $k = 102$, but $k$ cannot exceed $|C|$.
7. Considering the constraints and the problem's requirement for a solution, the minimal possible value of $|A \cap B \cap C|$ that fits the equation, despite the constraints, is found to be $101$.

Thus, the minimum possible value of $|A \cap B \cap C|$ is $\boxed{101}$.

