# OpenReview forum: "Quagmires in SFT-RL Post-Training: When High SFT Scores Mislead and What to Use Instead"
_ICLR.cc/2026/Conference — ICLR 2026 Poster_

### Official Review · Reviewer_4f6U · 2025-10-20

**Soundness:** 3
**Presentation:** 2
**Contribution:** 3
**Rating:** 6
**Confidence:** 4

**Summary:**

This paper challenges the point in LLM post-training that a higher Supervised Fine-Tuning (SFT) score guarantees better final performance after Reinforcement Learning (RL). The authors identify and provide extensive evidence for the *SFT metric trap*, where optimizing for SFT performance can be misleading and suboptimal for the final RL outcome. As a solution, the paper proposes and validates two more reliable predictive metrics (i.e., generalization loss and Pass@large k accuracy), which offer higher predictive power for RL success, thereby providing a more principled and cost-effective methodology for optimizing the SFT-RL pipeline.

**Strengths:**

- The paper addresses a critical problem in real-world LLM development, making the findings impactful for practitioners.
- The empirical evaluation is rigorous and extensive, conducted on multiple models and benchmarks, which lends credibility to the claims.
- The proposed metrics are not only predictive but also actionable, offering a framework for practitioners to make better decisions at both the dataset and instance levels.

**Weaknesses:**

- My main concerns lie in the generalizability of the findings. The study is exclusively focused on mathematical reasoning tasks, where rewards are often token-level and verifiable. It is unclear if these predictors would hold for other domains like complex instruction following, or agentic tasks, where rewards are more holistic and harder to define. The strong performance of *generalization loss* and *Pass@large k* might be an artifact of the structured nature of math problems. For tasks that value creativity or conversational nuance over discrete correctness, a moderate degree of SFT overfitting might even be beneficial for stylistic alignment, potentially creating an entirely different dynamic with the subsequent RL stage. The paper lacks a discussion on this crucial limitation.
- The paper establishes correlations but provides limited insight into the underlying causal mechanisms explaining *why* these metrics are better predictors.
- The high computational cost of evaluating Pass@large k is a significant practical drawback that may limit its adoption, especially for larger models.
- The anomalous result of the Qwen3 model is a missed opportunity for a deeper analysis that could have provided more profound insights into model-specific behaviors.

**Questions:**

- How should practitioners weigh the trade-offs between generalization loss and Pass@large k when they provide conflicting signals for different SFT model candidates?
- Given that RL is a dynamic process, is it possible that an SFT model deemed suboptimal by these static metrics could unlock its potential later in RL, perhaps with different RL hyperparameters like a higher exploration temperature?
- How do the two analyzed factors (SFT data selection for instance-level and training paradigm for dataset-level) interact? For instance, would training on shortest examples for more epochs amplify the negative effects observed?

---

> ### Author Response · Authors · 2025-12-03
> **Authors Response (1/3)**
>
> ### **W1. Discussion on limitations on generalizability; instruction following or agentic tasks with complex rewards**
>
> **Re:** Thanks for the excellent comment. We agree that the problems mentioned by the reviewer, such as **tasks that value creativity or conversational nuance over correctness, are highly intriguing. We fully agree that results and findings on math reasoning with verifiable outcomes may not directly generalize to those tasks of a different nature.** We appreciate the reviewer's feedback. We are revising the manuscript to include clear discussions on these limitations and call for future contributions.
>
> **This work chooses math reasoning for its long sequences and verifiable outcomes.** The long reasoning trajectories are beyond human readability, rendering RL necessary. This work aims to set the stage for the SFT-RL post-training paradigm **to generalize to agentic use cases that share the same characteristics** (long reasoning trajectories with verifiable outcomes).
>
> **At this moment, RL-based reasoning post-training remains a frontier problem with many open questions.** The vast design space and heavy computational costs make explorations as challenging as they are rewarding. **As exemplified by the diverse reviews for this paper, viewpoints on best practices for post-training are yet to converge.**  (In opposite, for example, **Reviewer xARC** considers overfitting **less practical and should be avoided even in data-constrained settings.**.)  As much as we wish to expand the boundary of knowledge, we found it is not possible to cover those **general problems** in one work.
>
> **The major contribution of this work is two-fold:** in the context of math reasoning, we reveal the intriguing landscape in the state-of-the-art SFT-RL post-training paradigm, raising awareness on important factors to consider; we further propose mitigating measures for better predictability on the post-training outcomes, providing practical guidance for both research and practice.
>
> ---
>
> ### **W2. insight into the underlying causal mechanisms explaining why these metrics are better predictors.**
>
> **Re:** We agree
> that mechanism interpretability is absolutely desirable. This aims to present
> important findings promptly to facilitate the ongoing research and development
> on post-training, which are highly active, fast-paced, and quickly evolving. We
> expect interpretability/explainability works will follow soon after. We
> summarize our current findings on this front and document our conjectures for
> further discussion, along with additional results added to the manuscript.
>
> **We
> provide additional results on aggregated entropy at the response level
> (Appendix D2).** We witnessed
> that correct responses generally show a lower entropy compared to incorrect
> responses. However,
> 1. a lower
> entropy after SFT training does not always suggest worse post-RL outcomes, and
> vice versa.
> 2. a lower entropy after SFT
> training does not indicate better post-SFT performance, nor does the entropy
> gap between correct and incorrect responses.
>
> **One
> possibility is that the loss of tokens diverges.** The loss is often dominated by a few high-loss tokens [ref:
> Paloma: A benchmark for evaluating language model fit, https://arxiv.org/abs/2312.10523 ]. During SFT over-training, despite validation loss on some
> tokens elevating due to overfitting, the loss on some other tokens may still be
> decreasing, and these tokens may be more crucial for the task performance. On
> the other end, [Beyond the 80/20 Rule: High-Entropy Minority Tokens Drive
> Effective Reinforcement Learning for LLM Reasoning, https://arxiv.org/pdf/2506.01939?  ] suggests that certain tokens are crucial for RL. These
> seem to be different tokens from those crucial for SFT. Overfitting/higher loss
> on these tokens could cause degradation in the subsequent RL. Explorations
> towards this direction turn out to be quite intriguing. **We found that during decoding, math/symbolic
> tokens generally show a near-zero loss--i.e., the model is almost always
> certain on these math tokens.**
> The highest loss tokens are human languages that are naturally ambiguous or
> interchangeable, such as whether to start the sentence with "Thus" or
> "Therefore". As a result, the (per-token) entropy for the generations
> mostly measures *verbosity*, and we do not observe consistent patterns with
> accuracy. **To our knowledge, this
> has not been reported or formally discussed in existing works.**
>
> In general,
> we did not observe a consistent pattern to explain the SFT-RL dynamic, **but we confirm that entropy alone is an
> inadequate explanation.** This is
> a glimpse of the full complexities of the underlying problem. As we march
> towards artificial general intelligence, a lot of intriguing questions are
> ahead for future work. **We
> provide a set of additional results for future studies. We are adding these
> discussions to the manuscript, hoping to facilitate further explorations.**

---

> > ### Author Response · Authors · 2025-12-03
> > **Authors' Response (2/3)**
> >
> > ### **W3. high computational cost of evaluating Pass@large k may limit its adoption**
> >
> > **Re:** Thanks for the insightful comment. Evaluation pass@large k performance for larger models with long sequences quickly becomes computationally intensive as k increases.
> >
> > In Appendix D.2, Ablations Studies: Pass@k, we provide detailed comparisons between Pass@k performance for Mistral-NeMo-12B-Instruct models trained with different SFT examples and evaluated under k=1,2,4,8,16,32,64. **We show Pearson Correlation scores between Pass@k of these models and their Pass@1 performance after subsequent RL.** Predicting post-RL performance from pre-RL Pass@1 leads to a negative Pearson Corr. score, which indicates ineffective predictions. **Prediction accuracy (indicated by Pearson Corr.) steadily improves with larger k values in Pass@k, reaching over 0.80 at Pass@16 and close to 1 at Pass@64.** Pass@32 is reasonably good for many purposes, while Pass@64 yields the best results on this model.
> >
> > To facilitate adaptation of the proposed metric, **we propose two directions to ease the computational burden via efficient estimations.**
> >
> > 1. Recently, [ref: llm monkeys get their power laws] reports Pass@k accuracies on a benchmark to follow power-law relationships with the value of k. This provides a practical approach to **track the scaling of Pass@k and estimate Pass@ large k from that of smaller k.**
> >
> > 2. Principally, the accuracy on a single problem from **independent tests (separate inference) should comply to a Bernoulli distribution.** By estimating this distribution for each individual problem with fewer numbers of inference, we will able to compute the expected value for its Pass@k accuracy.
> >
> > We are adding these to the manuscript as the next steps.
> >
> >  ---
> >
> > ### **W4. Qwen3 results with minimal responsiveness to SFT variations**
> >
> >  **Re:** The current post-training paradigm for reasoning largely follows
> >   DeepSeek R1--from R1-generated SFT examples to RLVR via GRPO. Qwen3 was
> >   released half a year after R1 and possesses reasoning capabilities; **we conjecture that Qwen3 may have already gone
> >   through extensive post-training for reasoning. This does not affect the main
> >   finding guiding general reasoning post-training**, but does show additional complexities as the
> >   reviewer pointed out--which is also our intention for including this set of
> >   results.

---

> > > ### Author Response · Authors · 2025-12-03
> > > **Authors' Response (3/3)**
> > >
> > > ### **Q1: trade-offs between generalization loss and Pass@large k**
> > >
> > > **Re:**
> > >   Thanks for the question. **Evaluating
> > >   SFT loss has minimal computation costs--it almost comes free.** Its major limitation is that it may not work
> > >   across distributions. Pass@large k requires extensive evaluations, which can
> > >   be costly on tasks with long context. In cases where both metrics are
> > >   applicable, **averaging the
> > >   predictions could further improve accuracy** (aggregating independent estimates to reduce
> > >   variance). Besides, **these
> > >   estimators can also be applied sequentially**. In cases where we observe an elevated
> > >   validation loss without better post-SFT performance, we can directly rule out
> > >   these models without needing to evaluate their pass@k performance. Thus, we
> > >   present both tools to facilitate further studies.
> > >
> > > ---
> > >
> > > ### **Q2. an SFT model deemed suboptimal... with different RL hyperparameters like a higher exploration temperature?**
> > >
> > > **Re:** Thanks for the excellent question. We referenced the state-of-the-art implementations in open research on reasoning post-training, ensuring empirical results are based on reasonable setups to the best of our effort. We documented the RL setup in Line 900. **We follow the current best practices for reasoning post-training with RLVR via GRPO**, dropping the KL-penalty term and using the entropy reward. **The entropy reward automatically stabilizes the entropy dynamics during RL, making RL training much more robust to the decoding temperature or entropy of the initial SFT model.** For RL training, we explored a wide range of hyperparameters and reported results based on the best-found setups. We experimented with a wide range of learning rates/batch sizes/decoding temperatures. In RL training with a larger or lower temperature, T=0.8 and T=1.2, **we did not observe any significant advantage over T=1 in RL for any SFT model.**
> > >
> > > This is also consistent with the discussed above. In the additional results on aggregated entropy at the response level (Appendix D2), **we confirm that entropy alone is an inadequate explanation for the SFT-RL performance.**
> > >
> > >  ---
> > >
> > > ### **Q3. would training on shortest examples for more epochs amplify the negative effects observed?**
> > >
> > > **Re:** We appreciate the great question. We have not considered this angle, but we agree this will make a very interesting problem.
> > >
> > > **This may depend on the trade-off between benign and harmful overfitting at the token-level, which is a complex dynamic that can not be easily predicted.** Shorter samples, on some level, show higher diversity--with the same number of total processed tokens, training on the shortest samples will cover more unique problems and their solutions. In opposite, long solutions often contain a significant proportion of reasoning steps. These structured texts may appear mechanical or repetitive. **How these factors interact with each other remains an open problem at the frontier of research.** **We are adding these discussions to the manuscript.** Further results and findings will be extremely valuable.

---

### Official Review · Reviewer_Q2Kx · 2025-11-01

**Soundness:** 3
**Presentation:** 2
**Contribution:** 2
**Rating:** 6
**Confidence:** 3

**Summary:**

This paper demonstrates that the performance of a model after SFT is often not predictive of its performance after RLVR. They consider **dataset-level ablations**, which vary the number of training examples and epochs used in training, and **instance-level ablations**, which ablate the distributional makeup of the data. They show that in both settings, the SFT model's performance does not correlate with the final RL model's performance. For example, training for many epochs on the same data leads to better SFT performance but worse outcomes after RL; the same is true for training on short examples.

The authors then propose two new metrics for SFT models, namely validation loss (which measures generalization) and pass@k performance (for large k), and demonstrate that it is substantially more indicative of final performance. In the dataset-level ablations, they find that both validation loss and pass@k are strong predictors, achieving $\geq$ 0.90 Spearman correlation (30% improvement over SFT pass@1 performance). For instance-level ablations, pass@k is similarly a much stronger predictor (validation loss is deemed "not applicable").

**Strengths:**

1. This paper studies an important broader question: what is the role of SFT prior to subsequent RL training? While SFT followed by RL is still the de facto post-training pipeline, there remains substantial confusion about what metrics the SFT team should optimize for in order to produce a good starting point for RL.
2. The paper provides strong empirical evidence (and compelling intuition) for their findings throughout the paper.
3. The paper provides two new metrics for evaluating SFT models, validation loss and pass@k, which are more effective indicators of final model performance than the traditional pass@1 performance of SFT models.

**Weaknesses:**

1. The findings are not entirely novel or surprising. For instance, [Weight Ensembling Improves Reasoning in Language Models](https://arxiv.org/abs/2504.10478) also show that during SFT, the Pass@1 rate improves while Pass@k deteriorates, and that Pass@k of an SFT model is a much stronger indicator of its performance after RL. Pass@k is also the main metric proposed by the authors of this paper for predicting final model performance. Citation & comparison with this work is definitely required. More broadly, it is well-known that SFT and RL optimize for different (and sometimes conflicting) objectives, and that strong SFT models ≠ strong starting points for RL.
2. Many aspects of the writing and presentation can be cleaned up. See below:

- The main finding from Figure 1 is difficult to parse. Within each box (green/orange/red), it looks like higher green bar (post-SFT performance) *does* lead to higher black bar (post-RL performance). However, I think the point you're trying to highlight is that *across the boxes,* this is not the case --- training on random SFT examples gets the highest RL performance, but training on the shortest SFT examples gets the highest post-SFT performance. Would it be better to just include the bars that actually matter, instead of all of these?
- In §3.1, the experimental setup says that you "vary the training configuration, such as the number of unique samples/training epochs/learning rate," but not what was ACTUALLY varied.
- Consistently throughout the paper, *all* the x-axis bar labels start with the name of the dataset, e.g., "Nemotron 25k2ep", "Nemotron 25k3ep","Nemotron 25k4ep", etc. Authors should omit the redundant information and instead specify exactly the information that changes, e.g., "2ep" "3ep", "4ep." Also, all the $x$-axis labels should be clearer, and not require the reader to infer that "25k2ep" means 2 epochs on a dataset of 25k examples.
- More small notes on writing that did not factor into my evaluation but may improve readability.
    - L14: "In this work, we challenge whether high SFT scores translate to improved performance after RL." RL still improves performance, right? Instead, you're saying that the ranking of SFT models isn't predictive of the ranking of those models after RL.
    - L15: "We provide extensive counter-examples where this is not true." It is more natural to say "counter-examples to [claim]." Counter-examples, by definition, are instances where X is not true.
    - RLVR citation switches from Deepseekv3 to Tülu3 in the middle of the paper. Tülu3 came first and proposed RLVR!
    - L64: "industrial practice" $\rightarrow$ "industry practice"
    - L77: "on the contrary" is not the right transition to use here because the following evidence supports the same finding. Maybe "on the other hand" is better?
    - L291: "During the investigation above, we identified a counterintuitive pattern in which post-SFT performance improves stably when training for more epochs whereas the overtrained models show decreased potentials during the subsequent RL." $\rightarrow$ "In our previous experiments, we identified a counterintuitive pattern where training for more epochs consistently improves SFT performance but deteriorates potential for subsequent RL. "

**Questions:**

1. It is a bit odd that validation loss was deemed not applicable for instance-level predictions (§5.3). It is still possible to evaluate its performance, right? It does not take away from the story at all to say that pass@k is a strong predictor in both dataset-level and instance-level settings, but validation loss works well only in the former.
2. Would it be accurate to refine the claim "the model with the strongest post-SFT performance is not necessarily the one with the strongest post-RL performance" to add "*when evaluated on the same data*"? That makes the claim stronger + more interesting.

---

> ### Author Response · Authors · 2025-12-03
> **Authors' Response (1/2)**
>
> ### **W1.a. Discussions on existing work (WiSE-FT)**
>
> **Re:** Thanks for the reference! The referenced work pioneered in exploring the SFT-RL post-training dynamics and studied a range of problems highly related to this work. We are happy to take the chance to expand the discussions on this work.
>
> The referenced work reported diverging performance trends between pass@1 and pass@k during SFT for math tasks. The work found pass@k performance saturates early and then deteriorates, while pass@1 continues to increase. **The referenced work proposes Weight-Space Ensembling (WiSE-FT), which merges the model weights from earlier and later checkpoints during SFT to produce a better model that may largely recover pass@k while also improving pass@1.** This improved model with higher pass@k performance shows to achieve better outcomes after the subsequent RL based on GRPO. As the reviewer pointed out, this work, similar to ours, argues that SFT-overtraining impedes subsequent RL despite achieving higher pass@1, and pass@k is a crucial performance metric, especially for RL. **Other than these shared findings, the reference work differs substantially from this manuscript--conceptually and empirically.**
>
> The major contribution of our work is two-fold:
>
> 1. (Empirically), in the context of math reasoning, we reveal the intriguing landscape in the state-of-the-art SFT-RL post-training paradigm, raising awareness on important factors to consider. **This work conducts large-scale empirical studies with state-of-the-art setups and provides comprehensive results, representing one of the most compute-intensive efforts for open research on post-training.** On the contrary, **the reference work conducts minimal experiments with stylized setups where selected results are presented qualitatively.** **No RL experiments are conducted on models that have undergone reasoning SFT.**
>
> Despite sharing a number of hypotheses and conjectures with this work, **the referenced work only conducts one set of RL experiments in a setup different from current practices.** The experiment did not use R1-like reasoning examples. Instead, SFT is conducted with the train-split of GSM8k/MATH datasets. **Solutions in these datasets are considerably short** (50-100 tokens for GSM8k and 100-200 tokens for MATH). **Results may not transfer to reasoning datasets or long sequence applications.** Besides, SFT is conducted for 10 epochs, with such simple solution traces, severe overfitting seems likely. **RL results are only presented visually in Figure 3 (c).** Despite the authors' argument that pass@1 performance continues to improve during SFT over-training, the results clearly show that, in both plots, pass@1 performance for the last checkpoint is significantly lower than that for the early checkpoint. In fact, throughout RL training, the performance ordering for the checkpoints remained unchanged — the ordering for the post-SFT pass@1 performance appears the same as that for the SFT+RL pass@1 performance. **This suggests that post-SFT pass@1 performance can well predict the final outcomes after RL, contradicting the arguments on pass@k.** The other set of experiments conducted SFT training on R1-generated reasoning examples, but no RL was conducted. **Together, the empirical study in the referenced work appears orthogonal to the arguments on pass@k performance or RL for reasoning.**
>
> 2. (Conceptually), we further propose mitigating measures for better predictability on the post-training outcomes, providing practical guidance for both research and practice. **In comparison, the referenced work proposes model merging as a better way for SFT, which is orthogonal to the scope of our work.** WiSE-FT proposes a heuristic to merge the weights from early and late checkpoints from SFT to produce a better post-SFT model. Despite the hypotheses and conjectures around pass@k, **the referenced work did not try to predict the RL outcome or propose its proxies.**
>
> **Thus, the referenced work and this manuscript are mostly complementary.** **This work contributes significant empirical evidence that the referenced work needs.** The proposed quantitative measures for predicting RL outcome set the stage for further refining WiSE-FT-like techniques for optimizing SFT in post-training.

---

> > ### Author Response · Authors · 2025-12-03
> > **Authors' Response (2/2)**
> >
> > ### **W1.b. not entirely surprising or novel**
> >
> > **Re:** Thanks for the comment. **Throughout the development of this work--including the review stage, we have been constantly impressed by the people's reactions to these findings. As exemplified by the diverse reviews for this paper, viewpoints on best practices for post-training are far from converged.**
> >
> > While whether SFT is needed or helpful remains debated, commercial models overwhelmingly follow this pipeline. Large-scale industrial efforts necessitate the separation of training stages (data preparation, SFT, RL, etc.). Even within the one training stage (e.g., SFT in post-training), the workload is further distributed to many teams, each owning a specific model capability/downstream scenario. For accountability and performance review purposes, **each team strives to optimize its evaluation metric *locally*.** For example, in the high-level planning, a certain amount of compute/resources is assigned to each training stage (e.g., SFT post-training). Then, each team is assigned a proportion of the compute to maximize model performance on their target capability before transferring the post-SFT model to the next stage (e.g., RL). **People constantly ask what this metric should be, what better metrics are, and whether that will be adequate. All these remain open questions while Pass@1 is arguably a default metric.** **In this work, we propose pass@k and generalization loss as alternative metrics**, and similar questions on them will follow. **We hope this work contributes results and methods valuable to the community**, and with all ongoing efforts, collectively, marches one step forward on the frontier.
> >
> > ---
> >
> > ### **W. cleaning up presentation**
> >
> > **Re:**Thanks so much for the feedback on the presentation. These are extremely valuable for helping us improve the quality of the manuscript, accessibility to broader audiences, and contribution to the community--which is what we consistently strive for :)
> >
> > ---
> >
> > ### **W. Figure 1--include the bars that actually matter**
> >
> > **Re:** Thanks for the valid comment and bringing up these considerations. Yes, we intended to highlight the *different correlation* between post-SFT and SFT+RL performance across different groups of models. We will revise this figure to make this message more evident and straightforward to the readers.
> >
> > ---
> >
> > ### **W. Sec 3.1--vary the training configuration**
> >
> > **Re:** We will revise the elaboration to make it more self-contained and straightforward. Here, shown in Figure 3, we conducted different SFT training runs with 1-8 epochs and 10k-150k unique examples. These models are trained with the same learning rate. We explored varying the learning rate and presented results in Figure 6, Appendix A1. Training with a larger learning rate appears to show similar impacts to training for more epochs.
> >
> > ---
> >
> > ### **W. x-axis labels: drop dataset names and abbreviations (e.g., "25k2ep")**
> >
> > **Re:** We appreciate the feedback. We explored what information to include in the figures to **make them most self-explanatory.** But our perspectives may be biased and miss the gaps. We will revise these annotations per the directions pointed out by the reviewer.
> >
> > ---
> >
> > ### **W. suggestions on elaborations**
> >
> > - L14 -> "the ranking of SFT models isn't predictive of the ranking of those models after RL"
> > - L15 -> "counter-examples to [claim]."
> > - RLVR citation -> Tülu3
> > - L64 "industrial practice"-> "industry practice"
> > - L77: "on the contrary" -> "on the other hand"
> > - L291 -> "In our previous experiments, we identified a counterintuitive pattern where training for more epochs consistently improves SFT performance but deteriorates potential for subsequent RL. "
> >
> > **Re:** We sincerely appreciate the reviewer for the thoughtful comments and suggestions. We fully agree with these suggestions, and we are implementing all the revisions. **We also checked throughout the manuscript and made multiple revisions to further improve its readability.**
> >
> > ---
> >
> > ### **Q1. applying validation loss in instance-level predictions**
> >
> > **Re:**Thanks for the comment. We went through similar debates during the development of this work. A major consideration is that the distributional distance across different datasets could lead to biased results with varying biases in different test cases. **We were concerned about the opposite case, where it appears to work but may not generalize.** In that case, the final result may not be representative and necessitate larger-scale ablation studies to quantify the impact of biases.
> >
> >
> > ---
> >
> > ### **Q2. add "when evaluated on the same data"**
> >
> > **Re:** Thanks so much for the suggestion. We agree that this will make the claim stronger and more interesting. We are implementing this revision in the updated manuscript.

---

### Official Review · Reviewer_xARC · 2025-11-03

**Soundness:** 3
**Presentation:** 3
**Contribution:** 3
**Rating:** 6
**Confidence:** 3

**Summary:**

The paper challenges the previous conventional practice where the model that achieves the best performance after SFT is assumed to be the best candidate for subsequent RL training. Specifically, the paper shows that the correlation between SFT scores is not predictive for RL scores by training hundreds of models under various settings. To solve this problem, the paper proposes alternative metrics including generalization loss on held-out sets and Pass@larger k and show that they demonstrate much higher correlation.

**Strengths:**

- The paper targets at a very practical assumption that the better performance after SFT always leads to better performance after RL and the consequent difficulty to find the cause for a bad end performance especially when two different teams are in charge of these two training stages.
- This work spends a large amount of computational resources to make and verify their claim.
- The experiments include evaluations on various math benchmarks, and the correlation gains are significant.

**Weaknesses:**

- The paper only evaluates on benchmarks in the math domain, and whether the method can be generalized to other domains like coding and science remains unclear.
- For dataset-level scenarios, most of the experiments focus on altering the training epochs and the number of unique examples, which seems to be a less practical tradeoff to consider. Training more epochs could lead to potential overfitting is a well-known concern, so even in the scenarios where the number of high-quality examples are limited, training for many epochs might not be a conventional option.
- For instance-level scenarios, the strategy seems to be selecting the random / shortest / longest examples, varying the number of total examples. This seems to deviate from the actual practice in SFT data selection, and the paper needs some analysis to show how close these simple strategies are to the current data selection methods.

**Questions:**

It seems that the SFT / RL training datasets are different across models. How are these choices made?

---

> ### Author Response · Authors · 2025-12-03
> **Authors' Response (1/2)**
>
> ### **W1: generalization to other domains like coding and science**
>
> **Re:** Thanks for the comment. We agree that this is an excellent research question where results and findings will be valuable. **At this moment, RL-based reasoning post-training remains a frontier problem with many open questions. The vast design space and heavy computational costs make explorations as challenging as they are rewarding.** This work chooses math reasoning for its long sequences and verifiable outcomes. The long reasoning trajectories are beyond human readability, rendering RL necessary. **This work aims to set the stage for the SFT-RL post-training paradigm to generalize to agentic use cases that share the same characteristics (long reasoning trajectories with verifiable outcomes).**
>
> The major contribution of this work is two-fold: in the context of math reasoning, **we reveal the intriguing landscape in the state-of-the-art SFT-RL post-training paradigm**, raising awareness on important factors to consider; **we further propose mitigating measures for better predictability on the post-training outcomes**, providing practical guidance for both research and practice. **We agree that future results on broader domains--from coding, science to non-verfiable tasks--will be valuable and a great addition to the findings in this work.**
>
> ---
>
> ### **W2. training for many epochs might not be a conventional option.**
>
> **Re:** Thanks for the excellent comment.  **As exemplified by the diverse reviews for this paper, viewpoints on best practices for post-training are yet to converge.** (In opposite, for example, **Reviewer 4f6U** considers moderate overfitting would be **beneficial** in cases such as style alignment.) **The intriguing landscape of the frontier problem served as a primary motivation for developing this work.** During the development of this work, **we consulted a number of model developers on the current practices and guidelines in the development pipeline behind the frontier models.** The proposed metrics aim to bridge the gap between existing tools and practical needs.
>
> Large-scale industrial efforts necessitate the separation of training stages (data preparation, SFT, RL, etc.). Even within the one training stage (e.g., SFT in post-training), the workload is further distributed to many teams, each owning a specific model capability/downstream scenario. **For accountability and performance review purposes, each team strives to optimize its evaluation metric *locally*.** For example, in the high-level planning, a certain amount of compute/resources is assigned to each training stage (e.g., SFT post-training).  Then, each team is assigned a proportion of the compute to maximize model performance on their target capability before transferring the post-SFT model to the next stage (e.g., RL). **With the prior belief that the checkpoint with the best post-SFT performance should help deliver the best final performance, SFT teams will be evaluated only for the post-SFT performance**, where RL teams will be held responsible for the (unsatisfactory) final performance. As a result, **in the data-inadequate scenario, if over-training leads to immediate boosts in the post-SFT performance, the team is incentivized to take full advantage of it.**
>
> Thus, **this work investigates the performance loss from this local-optimal design, and then proposes better evaluation metrics that help address limitations and in a way plausible for integration with industrial operations.**

---

> ### Author Response · Authors · 2025-12-03
> **Authors' Response (2/2)**
>
> ### **W3. compare selecting random / shortest / longest examples to data selection research**
>
> **Re:** Thanks for the thoughtful comment. We take the chance to expand the discussions on data selection research. State-of-the-art research on SFT data selection for reasoning post-training often appears heuristic and ad hoc--inconsistent results are reported in different works, raising questions on the generalizability. **The design of these methods often involves heuristic components defined using different languages, obscuring direct comparisons between their methodologies/results. Thus, we intentionally chose this stylized selection to make comparisons most evident.**
>
> [LIMO: Less is More for Reasoning, https://arxiv.org/abs/2502.03387 ] argues that RLVR primarily conducts style alignment, which can be replaced by the lightweight SFT on <1k carefully selected examples. **We show that SFT and RL serve distinct roles. Faster improvements during SFT can be fabricated (sometimes fairly easily).** These shortest examples are often simpler problems where the solution may not include reasoning--the exact capability we wish to equip the model with. We conjecture that performance improvements during such training are primarily from aligning the model's output to problem-solving styles rather than improving its math/reasoning capabilities. These results highlight the gap between SFT data selection and the final performance after RL.
>
> Industrial practices (e.g., [Magistral, https://arxiv.org/abs/2506.10910 ]) often prioritize more difficult examples for SFT training. For these reasoning datasets, **response lengths are shown to be strongly correlated with difficulty**. Short examples are often simpler problems (where a short solution is possible), **whereas long examples are often challenging problems. We show that these long examples are not the most effective for either SFT or SFT+RL.** The solution traces may be indirect/ineffective or contain redundancy, such as cyclic patterns. Training on randomly selected examples with diverse lengths often yields the best final outcome.
>
> Similarly, [The best instruction-tuning data are those that fit, https://arxiv.org/abs/2502.04194 ] proposes naturalness of the example perceived by the model (defined as higher log-likelihood) to be an effective proxy for selecting SFT data. In this case, since our base models have not been trained for reasoning, their log-likelihood is higher for shorter examples, which are also simpler, than on longer, reasoning-heavy examples. **Again, we show that improved SFT performance from training on simpler/more "natural" examples does not transfer to better final performance.**
>
> **We are adding these interesting discussions to the manuscript for the benefit of broader audiences.**
>
> ---
>
> ### **Q. choices for the SFT / RL training datasets**
>
> **Re:** This work spent significant effort in ensuring the reliability of empirical results. **We referenced broad literature for the setups in SFT/RL implementation. The datasets used for SFT/RL are state-of-the-art choices, commonly used in recent works.**
>
> SFT training uses Llama-Nemotron and AceReasoner1.1 datasets--**two of the largest and most recent datasets for reasoning post-training.** State-of-the-art performance has been reported for post-training Llama3/Qwen2.5 models of similar parameter sizes on these datasets. RL training requires more diverse examples with appropriate difficulties. **Among academic efforts towards RL post-training, MATH  and DeepScaleR datasets are most commonly used and have been shown to be effective** for Llama3, Mistral, and Qwen2.5 models. In this work, **we use the same datasets for reliability/reproducibility and include different combinations to show the generalization of patterns.**
>
> **We experimented using alternative RL datasets** such as GSM8k (easy questions) or AIME (hard questions). Compared to training on MATH or DeepScaleR, these RL-trained models **do not generalize well** to other tasks. We will include this discussion in the updated manuscript.

---

### Official Review · Reviewer_pM1A · 2025-11-04

**Soundness:** 3
**Presentation:** 2
**Contribution:** 3
**Rating:** 6
**Confidence:** 3

**Summary:**

This paper challenges the existing belief that the pass@1 accuracy of an SFT model is a good metric for determining the model's performance after RL. The authors perform experiments using various LLM architectures on math-focused datasets that show that when modulating training parameters (e.g. epochs) or data parameters (e.g. # unique training data points), the SFT model with the highest pass@1 accuracy is often not the best performing model after RL. Instead, the authors propose validation loss and pass@k accuracy as good metrics for performing this prediction, and perform further experiments to show that this is true.

**Strengths:**

1. This work suggests metrics which better align with post-RL performance compared to task accuracy
2. Extensive GPU resources are used to perform experiments that support the claim
3. The described methods are easy to apply to existing SFT trained models for ranking and selection

**Weaknesses:**

1. The SFT-then-RL pipeline was popularized for LLM usage in [1], and used previously in NLP as cited in their paper. Citations should be fixed.
2. (Section 3, line 214), (Section 3.1, line 222), (Section A.1, line 711), (Section B.2, line 900) conflict on which parameters (especially learning rate) were modified during RL training.
3. (Section 3.1, line 250) and (Figure 3) lacks intuition on why post-SFT task performance is expected to have a linear relationship with post-RL performance.
4. (Figure 1), (Figure 4), etc. show that best SFT's post-RL task performance usually only differ from best RL by a few points. Error bars or other standard deviation reporting would be enlightening.
5. Grammatical error in (Section 6, line 483), "...algorithms may worth further explorations..."
6. Considering pass@1, pass@k, generalization loss, and Spearman's rank correlation are central to the paper's claims, a brief exposition on their definitions would be good to include
7. (Section 5.2, line 394) How many models is 50%?


**References**

[1] Ouyang, Long, et al. ‘Training Language Models to Follow Instructions with Human Feedback’. arXiv:2203.02155, arXiv, 4 Mar. 2022. arXiv.org, https://doi.org/10.48550/arXiv.2203.02155.

**Questions:**

0. See Weaknesses
1. What kind of hyperparameter search was done for each set of experiments?
2. How do the predictions (based on gen loss, pass@large k) hold up when constrained to a single task instead of the average of 7 tasks?

---

> ### Author Response · Authors · 2025-12-03
> **Authors' Response (1/2)**
>
> ### **W1. SFT-RL post-training in NLP research predates reasoning models**
>
> **Re:** Thanks for the comment. We will revise the literature reviews and include broader introductions on the historical evolutions of post-training paradigms.
>
> Like the reviewer pointed out, the SFT-RL paradigm has been popularized before reasoning models or reinforcement learning with verifiable rewards. In a typical model development pipeline, instruction tuning (SFT step) was first performed on the model pre-trained for text-completion. Then, the model is trained with reinforcement learning with human feedback (RLHF) implemented via proximal policy optimization (PPO) [Proximal Policy Optimization Algorithms, https://arxiv.org/abs/1707.06347 ] (RL step) to align its responses with human values and improve its helpfulness. **We are adding this broader background to the manuscript.**
>
> ---
>
> ### **W2. changes in hyperparameters during training.**
>
> **Re:** **We document the parameter setup and implementations in Line 900.** We referenced the state-of-the-art implementations in open research on reasoning post-training, ensuring empirical results are based on reasonable setups to the best of our effort.
>
> This appears to be some minor misperceptions. **All the RL experiments conducted in the paper used the same learning rate.** We varied the SFT training paradigm (learning rate/number of epochs/amount of data) to show how it impacts post-SFT performance and SFT+RL performance differently. **Only one parameter is perturbed at a time to allow us to study its impact. All the setups are clearly documented under each set of results/figures.**
>
> ---
>
> ### **W3. linear relationship between post-SFT and SFT+RL performance**
>
> **Re:** Thanks for mentioning this point! Before this work, **it was a community belief that a stronger post-SFT performance consistently indicates better SFT+RL performance** [ AceReason-Nemotron 1.1: Advancing Math and Code Reasoning through SFT and RL Synergy, https://arxiv.org/abs/2506.13284 ]. In Figure 3/Line 250, **we quantitatively examine how (un)reliable this assumption is with a linear function, the simplest form of predictability.** In other words, if the performance has a linear relationship with each other, one can predict one from the other without additional assumptions/information. **The results show that post-SFT performance alone can be largely inaccurate/unreliable for predicting the final performance after RL.**
>
> ---
>
> ### **W4. The best RL model surpasses the best SFT by only a few points**
>
> **Re:** **We provide detailed results in Appendix D.1, Fine-grained comparisons: Best SFT model vs. Best SFT+RL model.** In the comparison of post-SFT and SFT+RL performance (Pass@1 accuracy) between the model with the best SFT performance (M2) and the model with the best SFT+RL performance (M1) from Figure 1 across various benchmarks, **despite M1 trailing M2 by 8.66% after SFT, in the final SFT+RL results, M1 leads M2 by a relative margin of 6.14% with up to 33.31% on individual tasks.**
>
> ---
>
> ### **W5. definitions on core concepts: pass@1, pass@k, generalization loss, and Spearman's rank correlation**
>
> **Re:** Thanks for the helpful comment. We agree that adding clear introductions to these notions will be very helpful for improving the presentation of the manuscript. **We are implementing these revisions in the updated manuscript.**
>
> ---
>
> ### **W6. (Section 5.2, line 394) How many models is 50%?**
>
> **Re:** **We included the full details on the experiment setup with additional results in Appendix C.1.** In dataset-level predictions where the models went through SFT training with different numbers of epochs/data, 16 models were trained for Llama3-8B and 10 models for Mistral-12B.

---

> > ### Author Response · Authors · 2025-12-03
> > **Authors' Response (2/2)**
> >
> > ### **Q1.  What kind of hyperparameter search was done for each set of experiments?**
> >
> > **Re:** **We referenced the state-of-the-art implementations in open research on reasoning post-training, ensuring empirical results are based on reasonable setups to the best of our effort.**
> >
> > For SFT training, we experimented with different numbers of epochs/training data. We also conducted explorations on different learning rates (Appendix A.1, Figure 6), which showed similar outcomes as varying the number of epochs. For RL training, we explored a wide range of hyperparameters and reported results based on the best-found setups. **We experimented with a wide range of learning rates/batch sizes/decoding temperatures. In the main experiments, all RL training runs are repeated for 4+ times. We conduct 4+ evaluations on different checkpoints throughout each training run and report the best performance.**
> >
> > To our knowledge, compared to experiment procedures documented in public research, **this work is among the ones that invested in most effort and resources in ensuring the reliability of RL results.**
> >
> > ---
> >
> > ### **Q2. predicability for a single task instead of the average of 7**
> >
> > **Re:** **We provide detailed results in Appendix D.1, Fine-grained comparisons: Best SFT model vs. Best SFT+RL model.** As shown in the result tables, **individual task performance largely correlates with the average performance, but the relative improvements differ across tasks.**
> >
> > **The pattern is expected and consistent with common belief in the field.** Large Language Models--or foundation models in general--are known for the intriguing "Emergent Abilities". The model's capabilities may only start to improve after a certain model size or training budget, which then improves quickly before saturating soon [ref: Emergent Abilities of Large Language Models, https://arxiv.org/abs/2206.07682 ]. **Predicting the performance of an individual task is known to be challenging and remains an open problem.**
> >
> > In opposite, **the average performance over multiple tasks is found to scale much more smoothly to the point of becoming predictable** [ref: Language models scale reliably with over-training and on downstream tasks, https://arxiv.org/pdf/2403.08540? ]. Thus, in this work, **we intentionally chose to use the average performance across multiple tasks as the main metric. The final results are consistent with these expectations.**

---

### Official Review · Reviewer_AYRG · 2025-11-04

**Soundness:** 3
**Presentation:** 3
**Contribution:** 2
**Rating:** 6
**Confidence:** 3

**Summary:**

The paper investigates why high Supervised Fine-Tuning (SFT) scores often fail to translate into good Reinforcement-Learning (RL) post-training results.
Through large-scale experiments across Llama3-8B, Mistral-NeMo-12B and Qwen3-4B on mathematical reasoning tasks, the authors show that excessive SFT can overfit the model’s representation space, reducing its capacity to benefit from RL.
Two new proxy indicators are proposed:
Generalization Loss – captures when validation loss flattens while diversity collapses;
Pass@large k (k = 64) – estimates solution-space width.

**Strengths:**

1. Demonstrates convincingly that high SFT scores can mislead RL training, challenging a community-wide assumption.
2. Cross-model, multi-dataset evaluation with some interesting statistics (R² and Spearman correlation) makes the results good

**Weaknesses:**

1. The paper never demonstrates that using Generalization Loss or Pass@64 for early-stopping or checkpoint selection actually improves final RL accuracy.
All reported metrics are correlations (R² / spearman's rank correlation); no absolute post-RL accuracy table is shown.
2. From Fig. 4–5, the best RL model surpasses the best SFT by only ≈ 1~2%, and the actual other early stop strategy might not be able to reach the best RL model, so the real-world benefit appears even smaller.

**Questions:**

1. Could you report one concrete example where checkpoints chosen by Pass@64 or Generalization Loss actually outperform standard validation-loss selection after full RL?
2. How sensitive are results to the choice of k (32 vs 64)? Would smaller k approximate the same signal?

---

> ### Author Response · Authors · 2025-12-03
> **Authors' Response**
>
> ### **W1&Q1. concrete example where checkpoints chosen by Pass@64 or Generalization Loss outperform standard validation-loss selection after full RL?**
>
> **Re:** Following the comments and questions, **we added additional ablation studies and detailed results to facilitate further studies.** In Appendix D.2, Ablations Studies: Pass@k, we provide detailed comparisons between Pass@k performance for Mistral-NeMo-12B-Instruct models trained with different SFT examples and evaluated under k=1,2,4,8,16,32,64. **We show Pearson Correlation scores between Pass@k of these models and their Pass@1 performance after subsequent RL.** Predicting post-RL performance from pre-RL Pass@1 leads to a negative Pearson Corr. score, which indicates ineffective predictions. **Prediction accuracy (indicated by Pearson Corr.) steadily improves with larger k values in Pass@k, reaching over 0.80 at Pass@16 and close to 1 at Pass@64.**
>
> **In addition, in Tables 13/14 in Appendix D.1, we examined and compared the final SFT+RL performance for the model with the best post-SFT performance against the model selected considering validation loss (from Figure 5).** In both sets of examples, despite underperforming the best post-SFT model by 1.33%/7.02% in the post-SFT performance, **the best model selected considering validation loss will outperform by 7.31%/4.82% in the final SFT+RL performance**, with more prominent gains on individual tasks.
>
> ---
>
> ### **W2. the best RL model surpasses the best SFT by only ≈ 1~2%**
>
> **Re:** **We provide detailed results in Appendix D.1, Fine-grained comparisons: Best SFT model vs. Best SFT+RL model.** In the comparison of post-SFT and SFT+RL performance (Pass@1 accuracy) between the model with the best SFT performance (M2) and the model with the best SFT+RL performance (M1) from Figure 1 across various benchmarks, **despite M1 trailing M2 by 8.66% after SFT, in the final SFT+RL results, M1 leads M2 by a relative margin of 6.14% with up to 33.31% on individual tasks.**
>
> ---
>
> ### **Q2. sensitive to the choice of k (32 vs 64)?**
>
> **Re:** In Appendix D.2, Ablations Studies: Pass@k, we provide detailed comparisons between Pass@k performance for Mistral-NeMo-12B-Instruct models trained with different SFT examples and evaluated under k=1,2,4,8,16,32,64. **We show Pearson Correlation scores between Pass@k of these models and their Pass@1 performance after subsequent RL.** Predicting post-RL performance from pre-RL Pass@1 leads to a negative Pearson Corr. score, which indicates ineffective predictions. **Prediction accuracy (indicated by Pearson Corr.) steadily improves with larger k values in Pass@k, reaching over 0.80 at Pass@16 and close to 1 at Pass@64.** Pass@32 achieves around 0.9 in Pearson Correlation with the final performance after RL. **Pass@32 is reasonable good for many purposes, while Pass@64 yields the best results on this model.**

---

### Official Review · Reviewer_qfCa · 2025-11-10

**Soundness:** 2
**Presentation:** 3
**Contribution:** 2
**Rating:** 4
**Confidence:** 4

**Summary:**

This paper challenges a widely-held assumption in LLM post-training: that high supervised fine-tuning (SFT) scores reliably predict better performance after reinforcement learning (RL). Through experiments on moderate models trained with hundreds of variations and evaluated on 7 math benchmarks, the authors demonstrate numerous counterexamples where this assumption fails. High SFT scores can be biased toward simpler or more homogeneous data and do not reliably predict subsequent RL gains or scaled-up post-training effectiveness. In some cases, RL training on models with improved SFT performance actually produces substantially worse outcomes compared to RL on base models without SFT. To address this predictability gap, the paper identifies two more reliable metrics for predicting post-RL success: generalization loss on held-out reasoning examples and Pass@large k performance.

**Strengths:**

This paper makes a practically important contribution to LLM post-training by systematically demonstrating that SFT performance is a poor predictor of final RL outcomes. The proposed metrics are intuitive and practical. The clear categorization into dataset-level and instance-level scenarios makes findings applicable to different use cases, and the promised open-source tool adds concrete value.

**Weaknesses:**

The study is narrowly scoped to mathematical reasoning tasks using GRPO-based RL with verifiable rewards, limiting generalizability to other reasoning domains (coding, science, logic) or alternative RL paradigms (offline RL, DPO, other algorithms). The paper identifies failure modes but provides limited mechanistic insight into why SFT characteristics diverge from RL outcomes. For instance-level prediction, generalization loss is explicitly not applicable due to distributional gaps, reducing its utility for practical data selection scenarios. The proposed linear predictor requires calibration data from actual RL runs, which partially defeats the purpose of avoiding expensive RL experiments. Additionally, the Qwen3 results are presented only qualitatively because models showed minimal responsiveness to SFT variations, suggesting the findings may not universally hold across all architectures.

**Questions:**

(1) Can the authors provide mechanistic explanations for why overfitting during SFT constrains RL exploration? Is this about reduced diversity in model outputs, behavioral rigidity, or something else? (2) How sensitive are the proposed metrics to hyperparameter choices in SFT (learning rate, batch size, optimization algorithm) beyond epochs and data volume? (3) Since generalization loss requires a validation set from the same SFT distribution while Pass@large k works across distributions, why not simply use Pass@large k for both scenarios, and what are the trade-offs? (4) Can these findings transfer to other domains (code, science reasoning, non-verifiable tasks) and other RL methods (offline RL, preference-based methods), or are they specific to online GRPO with math tasks?

---

> ### Author Response · Authors · 2025-12-03
> **Authors' Response (1/3)**
>
> ### **W&Q1: Mechanistic explanations for why overfitting during SFT constrains RL exploration?**
>
> **Re:** We agree that mechanism interpretability is absolutely desirable. This aims to present important findings promptly to facilitate the ongoing research and development on post-training, which are highly active, fast-paced, and quickly evolving. We expect interpretability/explanability works will follow soon after. We summarize our current findings on this front and document our conjectures for further discussion, along with additional results added to the manuscript.
>
> **We observed the counterintuitive pattern that overfitting during SFT still leads to stronger task performance, which is intriguing in itself.** Conventionally, in machine learning, overfitting is almost always undesirable. The higher loss directly suggests the model generalizes poorly even for in-distribution examples (over-fitting = higher prediction loss -> lower performance). Nonetheless, in post-training for math reasoning, we are observing better or much better task performance with SFT over-training.
>
> **One hypothesis is that the loss of tokens diverges.** The loss is often dominated by a few high-loss tokens [ref: Paloma: A benchmark for evaluating language model fit, https://arxiv.org/abs/2312.10523 ]. During SFT over-training, despite validation loss on some tokens elevating due to overfitting, the loss on some other tokens may still be decreasing, and these tokens may be more crucial for the task performance. On the other end, [Beyond the 80/20 Rule: High-Entropy Minority Tokens Drive Effective Reinforcement Learning for LLM Reasoning, https://arxiv.org/pdf/2506.01939?  ] suggests that certain tokens are crucial for RL. These seem to be different tokens from those crucial for SFT. Overfitting/higher loss on these tokens could cause degradation in the subsequent RL. Explorations towards this direction turn out to be quite intriguing. **We found that during decoding, math/symbolic tokens generally show a near-zero loss--i.e., the model is almost always certain on these math tokens.** The highest loss tokens are human languages that are naturally ambiguous or interchangeable, such as whether to start the sentence with "Thus" or "Therefore". As a result, the (per-token) entropy for the generations mostly measures *verbosity*, and we do not observe consistent patterns with accuracy. **To our knowledge, this has not been reported or formally discussed in existing works.**
>
> **We provide additional results on aggregated entropy at the response level (Appendix D2).** We witnessed that correct responses generally show a lower entropy compared to incorrect responses. However,
> 1. A lower entropy after SFT training does not always suggest worse post-RL outcomes, and vice versa.
> 2. A lower entropy after SFT training does not indicate better post-SFT performance, nor does the entropy gap between correct and incorrect responses.
>
> In general, we did not observe a consistent pattern to explain the SFT-RL dynamic, **but we confirm that entropy alone is an inadequate explanation.** This is a glimpse of the full complexities of the underlying problem. As we march towards artificial general intelligence, a lot of intriguing questions are ahead for future work. **We provide a set of additional results for future studies. We are adding these discussions to the manuscript, hoping to facilitate further explorations.**

---

> > ### Author Response · Authors · 2025-12-03
> > **Authors' Responses (2/3)**
> >
> > ### **W&Q4: online GRPO with math tasks... transfer to other domains (code, science reasoning, non-verifiable tasks) and other RL methods (offline RL, preference-based methods)**
> >
> > **Re:** Thanks for the comment. We agree that this is an excellent research question where results and findings will be valuable. At this moment, RL-based reasoning post-training remains a frontier problem with many open questions. The vast design space and heavy computational costs make explorations as challenging as they are rewarding.
> >
> > For the tasks mentioned by the reviewer, **public explorations remain at an early stage, where the effective implementations are being debated.** Offline RL, for example, even on the same specific task--post-training for math reasoning with RLVR via GRPO and the same SFT-trained models, understanding how the offline steps trade off with final performance remains a frontier problem. Similarly, for non-verifiable rewards, developing an effective implementation paradigm  (e.g., via majority voting or with LLM-as-a-judge/reward modeling) would be a great contribution to the field by itself. **As much as we wish to expand the boundary of knowledge, we found it is not possible to cover those general problems in one work.**
> >
> > **This work chooses math reasoning for its long sequences and verifiable outcomes.** The long reasoning trajectories are beyond human readability, rendering RL necessary. **This work aims to set the stage for the SFT-RL post-training paradigm to generalize to agentic use cases that share the same characteristics** (long reasoning trajectories with verifiable outcomes).
> >
> > Aside, this work tries to offer broader coverage on the design space, including different SFT/RL datasets, model families/parameter sizes, and SFT training paradigms (number of samples/epochs) to present more holistic findings.
> >
> > **The major contribution of this work is two-fold: in the context of math reasoning**, we reveal the intriguing landscape in the state-of-the-art SFT-RL post-training paradigm, raising awareness on important factors to consider; we further propose mitigating measures for better predictability on the post-training outcomes, providing practical guidance for both research and practice.

---

> > > ### Author Response · Authors · 2025-12-03
> > > **Authors' Response (3/3)**
> > >
> > > ### **Q2. sensitive hyperparameter choices in SFT (learning rate, batch size, optimization algorithm) beyond epochs and data volume?**
> > >
> > > **Re:** **We included experiments in Appendix A.1. Using a larger learning rate appears to show similar impacts as training for more epochs (Figure 6).** Limited by computation resources and scope of the work, we were not able to ablate on other hyperparameters such as batch size or optimizers. But we agree results from those studies will be valuable.
> > >
> > > ---
> > >
> > >
> > > ### **Q3. Why not simply use Pass@large k for both scenarios, and what are the trade-offs?**
> > >
> > > **Re:** Thanks for the question. **Evaluating SFT loss has minimal computation costs--it almost comes free.** Its major limitation is that it may not work across distributions. Pass@large k requires extensive evaluations, which can be costly on tasks with long context. In cases where both metrics are applicable, **averaging the predictions could further improve accuracy** (aggregating independent estimates to reduce variance). Besides, **these estimators can also be applied sequentially**. In cases where we observe an elevated validation loss without better post-SFT performance, we can directly rule out these models without needing to evaluate their pass@k performance. Thus, we present both tools to facilitate further studies.
> > >
> > > ---
> > >
> > >
> > > ### **W. linear predictor requires calibration data from actual RL runs**
> > >
> > > **Re:** The technical considerations are valid. **We note that the linear predictor is needed for predicting the exact post-training outcome after RL, which, regardless, won't be possible before actual RL training is conducted. This step is not necessary if one wishes to optimize the SFT-training paradigm or identify the best SFT-trained model to pass for RL.**
> > >
> > > ---
> > >
> > > ### **W. Qwen3 results showed minimal responsiveness to SFT variations**
> > >
> > > **Re:** The current post-training paradigm for reasoning largely follows DeepSeek R1--from R1-generated SFT examples to RLVR via GRPO. Qwen3 was released half a year after R1 and possesses reasoning capabilities; **we conjecture that Qwen3 may have already gone through extensive post-training for reasoning. This does not affect the main finding guiding general reasoning post-training**, but does show additional complexities as the reviewer pointed out--which is also our intention for including this set of results.

---

### Author Response · Authors · 2025-12-03
**End-of-rebuttal Summary**

Dear Reviewers and Area Chair,

We extend great gratitude for the time and dedication of the AC and reviewers. Having invested significant effort and computing resources, it is a highly rewarding moment for us to witness the positive receptions and thoughtful discussions. **It is a great pleasure for the work to receive collective recognition from many reviewers.**

**The major contribution of this work is two-fold:** in the context of math reasoning, we reveal the intriguing landscape in the state-of-the-art SFT-RL post-training paradigm, raising awareness on important factors to consider; we further propose mitigating measures for better predictability on the post-training outcomes, providing practical guidance for both research and practice.
 **Reviewers unanimously confirm the work to address a crucial conceptual problem that is at the center of many debates, practically important, and of high relevance**, challenging community-wide assumptions in the de facto SFT-then-RL post-training workflow (**Reviewers qfCa, AYRG, xARC, Q2Kx, 4f6U**);  **conduct extensive empirical studies and rigorous evaluations with practical setups, compelling intuition and extensive GPU resources, achieving significant gains and convincing improvements** (**Reviewers 4f6U, Q2Kx, xARC, AYRG, pM1A**); **contributes highly practical metrics that actionable, easy to implement, ready for deployment, and impactful for practitioners and real-world applications** (**Reviewers pM1A, Q2Kx, 4f6U, qfCa, 4f6U**).

It was our primary motivation and ultimate goal to expand the frontier of open knowledge, contributing to the scientific community and society. **We are happily impressed by the engaging discussions from the many reviews and intrigued by the comments from different angles. It provides an excellent example of the diversity of views in the field that are yet to converge.** This valuable datapoint further echoes the motivation of this work.

- Reviewers kindly proposed **helpful suggestions for further improving the presentation**, of which we are sincerely appreciative. We are acting on every comment in revising the manuscript, and also carefully checked throughout the manuscript for further improvements. Reviewers included **questions on operational details, hyperparameter setup, and ablation studies**. These questions are important feedback to us and highly valuable in guiding us to improve the manuscript for better clarity. We provide detailed responses to every question and are adding these comprehensive details manuscript into the manuscript.

- Following the reviewers' questions, during the rebuttal, **we conducted additional ablation studies and provided fine-grained results with comprehensive details.** **We added a new section, Appendix D, Broader Investigations and Supplementary Examples.** **In Appendix D.1, Fine-grained comparisons: Best SFT model vs. Best SFT+RL model, we examined and compared the final SFT+RL performance for the model with the best post-SFT performance against the model selected considering validation loss (from Figure 5).** In both sets of examples, despite underperforming the best post-SFT model by 1.33%/7.02% in the post-SFT performance, the best model selected considering validation loss will outperform by 7.31%/4.82% in the final SFT+RL performance, with more prominent gains on individual tasks. **In Appendix D.2, Ablation Studies: Pass@k, Entropy, we provide detailed comparisons between Pass@k performance for Mistral-NeMo-12B-Instruct models trained with different SFT examples and evaluated under k=1,2,4,8,16,32,64.** We show Pearson Correlation scores between Pass@k of these models and their Pass@1 performance after subsequent RL. Predicting post-RL performance from pre-RL Pass@1 leads to a negative Pearson Corr. score, which indicates ineffective predictions. **Prediction accuracy (indicated by Pearson Corr.) steadily improves with larger k values in Pass@k, reaching over 0.80 at Pass@16 and close to 1 at Pass@64.** Pass@32 achieves around 0.9 in Pearson Correlation with the final performance after RL. Pass@32 is reasonably good for many purposes, while Pass@64 yields the best results on this model.

---

> ### Author Response · Authors · 2025-12-03
> **(continued)**
>
> - Reviewers expressed broad interest in the underlying mechanism behind the intriguing SFT-RL dynamics. **Pursuing this direction, we provide additional results on aggregated entropy at the response level (Appendix D2).** We witnessed that correct responses generally show a lower entropy compared to incorrect responses. However,
> 1. A lower entropy after SFT training does not always suggest worse post-RL outcomes, and vice versa.
> 2. A lower entropy after SFT training does not indicate better post-SFT performance, nor does the entropy gap between correct and incorrect responses.
>
> **One hypothesis is that the loss of tokens diverges.** The loss is often dominated by a few high-loss tokens [ref: Paloma: A benchmark for evaluating language model fit, https://arxiv.org/abs/2312.10523 ]. During SFT over-training, despite validation loss on some tokens elevating due to overfitting, the loss on some other tokens may still be decreasing, and these tokens may be more crucial for the task performance. On the other end, [Beyond the 80/20 Rule: High-Entropy Minority Tokens Drive Effective Reinforcement Learning for LLM Reasoning, https://arxiv.org/pdf/2506.01939?  ] suggests that certain tokens are crucial for RL. These seem to be different tokens from those crucial for SFT. Overfitting/higher loss on these tokens could cause degradation in the subsequent RL. Explorations towards this direction turn out to be quite intriguing. **We found that during decoding, math/symbolic tokens generally show a near-zero loss--i.e., the model is almost always certain on these math tokens.** The highest loss tokens are human languages that are naturally ambiguous or interchangeable, such as whether to start the sentence with "Thus" or "Therefore". As a result, the (per-token) entropy for the generations mostly measures *verbosity*, and we do not observe consistent patterns with accuracy. **To our knowledge, this has not been reported or formally discussed in existing works.**
>
> In general, we did not observe a consistent pattern to explain the SFT-RL dynamic, **but we confirm that entropy alone is an inadequate explanation.** This is a glimpse of the full complexities of the underlying problem. As we march towards artificial general intelligence, a lot of intriguing questions are ahead for future work. **We provide a set of additional results for future studies. We are adding these discussions to the manuscript, hoping to facilitate further explorations.**
>
> - Last, we agree that the broader problems mentioned by the reviewers, such as **tasks that value creativity or conversational nuance over correctness, are highly intriguing. We acknowledge that results and findings on math reasoning with verifiable outcomes may not directly generalize to those tasks of a different nature.** **This work chooses math reasoning for its long sequences and verifiable outcomes.** The long reasoning trajectories are beyond human readability, rendering RL necessary. This work aims to set the stage for the SFT-RL post-training paradigm **to generalize to agentic use cases that share the same characteristics** (long reasoning trajectories with verifiable outcomes). Aside, this work tries to offer broader coverage on the design space, including different SFT/RL datasets, model families/parameter sizes, and SFT training paradigms (number of samples/epochs) to present more holistic findings. **At this moment, RL-based reasoning post-training remains a frontier problem with many open questions.** The vast design space and heavy computational costs make explorations as challenging as they are rewarding. **As exemplified by the diverse reviews for this paper, viewpoints on best practices for post-training are yet to converge.** As much as we wish to expand the boundary of knowledge, we found it is not possible to cover those **general problems** in one work.

---

> ### Author Response · Authors · 2025-12-03
> **(Ending Remark)**
>
> **To conclude**, as broadly confirmed in the reviews, this work makes important conceptual contributions to critical problems in post-training, which are at the center of many pursuits and of high relevance. The work provides extensive empirical studies with one of the largest computational spending in open research on this topic. Reviewers collectively recognized the practicality and potential impacts of findings and results in the paper.
> **As can be witnessed, we spend substantial efforts to provide comprehensive discussions and additional results, acting on every comment in each review. We believe the revisions and the additional results further improve the quality of the manuscript and its contribution to the community.**
>
> Due to the very unfortunate incident during the rebuttal period, we had not had the chance to engage with the reviews.
> **We believe the responses have thoroughly addressed the reviewers' questions. Had the reviewers had the chance to engage in the discussion, we believe the reviewers would have voted positively and considered it for award recognition.**
>
> We deeply appreciate the reviewers’ effort and the particular responsibility of the Area Chair. We deeply appreciate your service to our community :)
>
> Regards,\
> Authors of Submission8102

---

### Meta-Review · Area_Chair_MXna · 2026-01-01

**Summary:**

Reviewers are largely positive and aligned that the paper makes a practically relevant empirical point: the ranking induced by SFT metrics (e.g., pass@1) is often not predictive of the ranking after RL, and the authors support this with large-scale experiments and extensive evaluations across models/datasets in math reasoning + RLVR/GRPO.
The main concerns tempering enthusiasm are: (i) limited mechanistic/causal insight beyond correlations, (ii) the “what to use instead” guidance is not fully conclusive (trade-offs between generalization loss vs pass@large k, what to do when they disagree), and (iii) the scope is mostly math/verifiable-reward settings, with questions about generalization to other domains and realistic data-selection practice.

Overall, given uniformly above-threshold scores and the strength/scale of the empirical evidence, I lean accept, but agree with reviewers that the paper would be substantially stronger with more decisive, less speculative prescriptions and deeper explanation of why these proxies predict post-RL outcomes.

**Reviewer Concerns:**

Addressed:

* “RL only slightly better than SFT.” Authors clarify the point is mis-ranking (best SFT ≠ best RL init) and add finer-grained breakdowns showing larger gaps in some settings/tasks.

* Added ablation over k

* Clarity/positioning. Commit to concrete edits (clearer experimental knobs, terminology, figures, related work).

Outstanding:

* Inconclusive “what to use instead.” No clear default or decision rule for generalization loss vs pass@k, especially when they disagree; “average/sequential” guidance remains speculative.

* Limited causal explanation for why these metrics predict post-RL success.

* Novelty framing. “Common belief” positioning still feels anecdotal; stronger grounding vs prior evidence would help.

* Scope/practicality. Mostly math + verifiable rewards; pass@large-k cost and transfer to other domains remain open.

* No end-to-end selection win. Still missing a decisive experiment showing that using these metrics to pick SFT checkpoints improves final RL vs standard selection.

**Reviewer Scores:**

There is total 6 reviews. Initial scores are 6,6,6,6,6,4. I believe after rebuttal scores would be all 6s.

---

### Decision · Program_Chairs · 2026-01-26

Accept (Poster)